# Regulative Roles of Metabolic Plasticity Caused by Mitochondrial Oxidative Phosphorylation and Glycolysis on the Initiation and Progression of Tumorigenesis

**DOI:** 10.3390/ijms24087076

**Published:** 2023-04-11

**Authors:** Nan Niu, Jinfeng Ye, Zhangli Hu, Junbin Zhang, Yun Wang

**Affiliations:** 1Shenzhen Engineering Labortaory for Marine Algal Biotechnology, Longhua Innovation Institute for Biotechnology, College of Life Sciences and Oceanography, Lihu Campus of Shenzhen University, Shenzhen 518055, China; nn2023fly@szu.edu.cn (N.N.);; 2College of Physics and Optoelectronic Engineering, Canghai Campus of Shenzhen University, Shenzhen 518060, China

**Keywords:** tumorigenesis, glycolysis, mitochondrial oxidative phosphorylation, metabolic plasticity, malignant proliferation

## Abstract

One important feature of tumour development is the regulatory role of metabolic plasticity in maintaining the balance of mitochondrial oxidative phosphorylation and glycolysis in cancer cells. In recent years, the transition and/or function of metabolic phenotypes between mitochondrial oxidative phosphorylation and glycolysis in tumour cells have been extensively studied. In this review, we aimed to elucidate the characteristics of metabolic plasticity (emphasizing their effects, such as immune escape, angiogenesis migration, invasiveness, heterogeneity, adhesion, and phenotypic properties of cancers, among others) on tumour progression, including the initiation and progression phases. Thus, this article provides an overall understanding of the influence of abnormal metabolic remodeling on malignant proliferation and pathophysiological changes in carcinoma.

## 1. Introduction

The energy metabolism of normal cells is maintained in a balanced state. In the presence of oxygen, partial metabolites are transmitted through respiratory chain enzyme complexes I–IV on the inner membrane of mitochondria, and adenosine triphosphates (ATPs) are finally synthesized by oxidative phosphorylation of adenosine triphosphate synthase (ATPase) [1,2,3], which only occupies a small part of the whole cell energy supply. More glycolysis-related enzymes are activated when cells are exposed to ischaemia and hypoxia to maintain the cellular energy supply [4]. However, in the 1920s, Otto Warburg’s pioneering study demonstrated that, unlike normal cells, which produce energy by oxidative phosphorylation of mitochondria, tumour cells mainly produce energy through glycolysis, even if the oxygen supply is sufficient [5]. This abnormal metabolic status in tumour cells is called the Warburg Effect, which is the most significant metabolic difference between normal and tumour cells. The Warburg Effect is responsible for various characteristics associated with the occurrence and development of tumour cells, such as their phenotype, material metabolism, molecular regulation, proliferation, invasion, immune escape, and angiogenesis stimulation [6,7,8,9,10].

Typically, in the presence of oxygen, glycolysis produces only two ATPs per glucose molecule, which is much lower than the ~38 ATPs produced per glucose molecule via oxidative phosphorylation. Thus, for energy supply, mitochondrial oxidative phosphorylation, rather than glycolysis (usually enhanced in the latter phase of energy production) should be used to meet the complex and vigorous biological and pathological needs of tumour cells. Generally, the contribution of glycolysis to total ATPs in tumour cells ranges from 1% to 64% [11]. However, with more research data, it has been revealed that tumour cells still retain relatively complete mitochondria and the ability of oxidative phosphorylation.

Tumour cells with glycolysis could be stimulated to use mitochondrial oxidative phosphorylation under specific conditions, and the capacity of mitochondrial oxidation that tumour cells retain is not just to meet energy demand [12,13,14]. These adjusted actions are mainly induced by their abnormal phenotype, functional transformation, or other physical and chemical factors caused by internal and external stimulators [15,16]. Glycolysis and mitochondrial oxidative phosphorylation constitute a diversified and sophisticated metabolic phenotype of tumour cells called metabolic plasticity. The metabolic plasticity of tumour cells allows them to have strong adaptability and elasticity to cope with intracellular and extracellular environments, which makes them highly plastic and pervasive in all aspects of tumourigenesis and development [17,18,19], including their initiation and progression. However, it remains unknown how tumour cells coordinate their gene regulation to balance their glycolysis and oxidative phosphorylation (OXPHOS) and subsequently affect their occurrence and development by regulating metabolic plasticity. These topic questions are also the hotspot and difficulty of current research and bring opportunities and challenges for clinical treatment.

The tumourigenesis process involves several stages: initiation of tumourigenesis, uncontrolled proliferation and unprogrammed cell death, development of tumour microenvironment (TME), and gaining the ability to invade and migrate [20]. The progression of tumour cells following the initial stage is caused by a series of gene mutations or genomic changes, as well as disorders of the cell cycle and frustration of immune surveillance function, which causes cells to divide continuously, escape apoptosis, immune phagocytosis, and inhibit growth [21,22,23,24]. This uncontrolled cell state gradually accumulates, eventually resulting in tumourigenesis. At the initial stages of tumourigenesis, many metabolic pathways may play important regulatory roles in tumourigenesis and development [25,26,27]. More importantly, to meet the needs of vigorous growth, cancer cells need to adjust, integrate, and change their metabolic pattern, which involves many oncogenes [28,29]. For example, the loss of NAD-dependent protein deacetylase sirtuin-6 (SIRT6) can produce a powerful metabolic rearrangement that can induce tumourigenesis [27,30]. The overexpression of glucose transporter type 3 (GLUT3) in non-malignant breast cancer cells can enhance the activation of epidermal growth factor receptor (EGFR), MAP kinase-activated protein kinase (MAPK), RAC-beta serine/threonine-protein kinase (Akt), and other signaling pathways as well as improve glucose metabolism and the expression of related oncogenes, thereby increasing the risk of cell transformation [31].

Tumour cells with immortality and unlimited proliferation ability gradually replace normal cells in the surrounding environment, absorb all possible nutrients, promote angiogenesis, and further develop cancer cells. When the environment cannot meet its growth needs, tumour cells change the expression pattern of endogenous regulatory molecules and the cell phenotype, secrete invasion factors, affect the TME externally, penetrate the blood circulation, and spread outwards. In this process, tumour cells adjust their adaptive strategies through metabolic plasticity regulation, such as adjusting the absorption mode of nutrients, which is accompanied by the transformation of glycolysis to OXPHOS [32,33]. Another example is to adjust the tumour microenvironment by secreting lactic acid to allow the infiltrating immune cells to undergo metabolic rearrangement and eventually escape [34,35,36,37].

Subsequently, metastatic tumour cells travel through the blood and lymphatic circulation, escape the surveillance of immune cells in various ways, infiltrate the blood vessel wall, and then reach the second plantable area for reciprocal circulation. In this process, tumour cells can also affect their ability to metastasize and invade in many ways by remodelling the metabolic processes of glycolysis and OXPHOS, such as adjusting epithelial–mesenchymal transition(EMT)-related proteins, changing the composition of the extracellular matrix, and antagonizing apoptosis [38,39,40,41].

Therefore, during the entire stage of tumour cell development, even though some factors and pathways can affect the future trend of tumours, they directly or indirectly regulate metabolism, and the regulation of metabolic plasticity will also affect the vigorous growth and other needs of tumour cells [42]. The regulation of plasticity between glycolysis and mitochondrial oxidative phosphorylation is one of the basic characteristics of tumour cells [29,42]. Correspondingly, this metabolic plasticity also affects the performance of anticancer drugs. Thus, this review aims to summarize and discuss the characteristics and regulatory processes of metabolic plasticity between glycolysis and mitochondrial oxidative phosphorylation during tumour initiation and progression to provide a new vision for the treatment and understanding of tumourigenesis and development.

## 2. Metabolic Plasticity in the Initiation of Tumourigenesis

The regulation of metabolic plasticity, dependent on glycolysis and OXPHOS, initiates tumour progression through a variety of complex and hard-to-explain mechanisms for its impact on tumour initiation. Researchers have proposed different theories regarding the initiation of tumour cells, such as oncogene theory [43], mutation theory [43], cell fusion theory [44], and two-hit hypothesis [45]. Therefore, we have discussed different aspects of the influence of metabolic plasticity on tumour initiation. While Potter and Pierce proposed that tumours might result from the inhibition of stem cell differentiation and maturation [46,47], many researchers believe that tumours may originate from a single clone of stem cells, which have undergone malignant transformation and show increased aggressiveness. Proliferating and differentiating stem cells have a high probability of gene mutation [48,49], genomic instability [50,51], and abnormal cell fusion [52,53], which lead to tumorigenesis [54,55,56,57].

Therefore, more researchers believe that tumourigenesis is initiated by the development of stem cells. That is, tumour stem cells, also known as tumour-initiating cells (TICs), are self-renewal cells with multiple heterogeneities. Tumour stem cells can divide asymmetrically to produce daughter cells with different phenotypes. This led to some doubts regarding whether the metabolic phenotype of tumour depends on a single tumour cell or composed of tumour cells with different metabolic phenotypes. To study the metabolic plasticity mediated by glycolysis and mitochondrial oxidative phosphorylation in the initial stage of the tumour, we focused on the effect of metabolic plasticity on the tumourigenesis in stem cells.

### 2.1. The Importance of Glycolysis and OXPHOS in the Transformation of Stem Cells into Tumour Stem Cells

Later, researchers began to explore the mechanism of metabolic regulation of stem cell-derived tumour stem cells. Many reports have shown that resting stem cells have typical metabolic characteristics and are more prone to glycolysis [58,59]. For instance, the transcription factor homeobox protein Meis1 (MEIS1) in haematopoietic stem cells (HSC) can regulate hypoxia inducible factor-1α (HIF-1α) expression [60], which further controls the expression of many glycolysis-related molecules. Differentiated daughter cells derived from HSCs also rely on glycolysis to meet their metabolic needs [61]. Some pyruvate dehydrogenase kinase (PDK) isoforms can restrict pyruvate entry into mitochondria to control the resting state of long-term repopulating haematopoietic stem cells (LT-HSCs) [59,62,63]. These studies have provided insights into the role of glycolysis in stem cells.

However, it is unclear whether the metabolic characteristics of glycolysis in stem cells are retained after they transform into cancer cells. As previously reported, the energy source of some glioma stem cells (GSCs) primarily depends on glycolysis [64,65,66]. In addition, the activation of some glycolytic enzymes, rather than oncogenes, can cause tumourigenesis and transformation, providing favourable evidence for the metabolic influence on tumorigenesis [67,68,69]. Notably, the biological and biochemical processes of some transcription factors, such as Myc proto-oncogene protein (c-Myc) and Nuclear factor NF-kappa-B (NF-κB), are mainly dependent on glycolytic reactions [9,70,71,72]. These transcription factors can activate epithelial stem cell(ESC)-like transcription programs [73] and induce dedifferentiation of epithelial non-stem cells to obtain characteristics of intestinal stem cells (ISC) [74], which could further promote the tumourigenesis.

Previous studies have also revealed a possible connection between glycolysis and tumourigenesis in normal somatic cells, except for reports on glycolysis in the cancer process of stem cells. For example, mutations in KRAS proto-oncogene and GTPase(KRAS) exist in most tumours, such as lung cancer, colon cancer, and pancreatic cancer [75,76,77], and introduce mutated KRAS (KRAS^G12D^) into normal human breast cells, which can initiate breast cancer [78]. KRAS can also directly regulate the key enzymes of glycolysis, such as hexokinase (HK), phosphofructokinase-1 (PFK1), lactate dehydrogenase A (LDHA), and glucose transporter Type 1 (GLUT1), thus affecting the glycolysis and impacting cell canceration [77,79].

Therefore, can it be considered that the metabolic phenotype of the initial process of tumour cells is dominated by glycolysis? Alternatively, is glycolysis rather than OXPHOS involved in the initiation of tumour cells? Many researchers have used a more efficient and controllable method by artificially stimulating tumour initiation to study the metabolic plasticity at this stage and accurately explore its influence on tumour initiation. Using this strategy, researchers confirmed that the active cycling crypt base columnar (CBC) leucine-rich repeat-containing G protein-coupled receptor 5 positive (Lgr5^+^) intestinal stem cells are the main cause of colon epithelial cell carcinoma (CRC). On basis of this model, later studies found that, compared with the adjacent paneth cells, a high expression of Lgr5^+^ intestinal stem cells have high activity of mitochondrial oxidative phosphorylation [80,81,82].

Bonnay et al. reported that, upon an initiation of neural stem cell tumourigenesis, glutamine absorption was transformed into OXPHOS and increased NAD^+^ biosynthesis, which might be an important pathway for the initiation of neural stem cell tumourigenesis [83]. That is, OXPHOS produced by excessive mitochondrial fusion and increased oxidation reaction and NAD^+^ may initiate tumourigenesis [83,84]. This result is surprising because tumours are currently widely believed to be glycolytic. The importance of oxidative phosphorylation in different tumours has been reported in several studies [85,86]. For example, Transcription Factor A mitochondrial (TFAM) plays a key role in mitochondrial activity and gene transcription, and the deletion of TFAM leads to the loss of the self-renewal ability of stem cells and subsequent cell canceration [87]. Tumour stem cells from the brain, pancreas, lung, and ovary rely on oxidative phosphorylation [88,89,90,91,92,93].

However, tumourigenesis initiated by OXPHOS has created new problems. First, as mentioned above, researchers have found that molecules and events related to glycolysis are involved in tumour initiation. In addition, when stem cells develop into tumour stem cells, they not only have the ability of unlimited proliferation, but they also “evolve” a series of molecular mechanisms to escape death [94], which, under non-mechanical pressure, is vitally important. Although mitochondria are required to initiate cell death, they can also induce iron death and cell apoptosis [95,96]. Mitochondria are also the main source of reactive oxygen species (ROS) which causes cell damage [97]. Relevant reports have confirmed that the inhibition of the glycolytic pathway can induce apoptosis of GSCs [98,99], which is caused by ROS produced through OXPHOS [100]. Therefore, the maintenance of mitochondrial oxidative metabolism may not be conducive to tumour cells’ long-term growth and proliferation in complex and changeable internal and external environments [96]. Moreover, Bonnay’s study did not clarify how the increased oxidation reaction led to the immortality of tumour cells. They also did not completely exclude the high levels of tumour-expressed glycolytic enzymes. Furthermore, Lgr5^+^ ISCs have a lower level of pyruvate oxidation and a higher level of glycolysis [101], which suggests that glycolysis is highly related to tumour initiation. These results suggest that stem cell tumourigenesis is more complex than previously thought.

### 2.2. Implications and Significance of Research on Metabolic Plasticity of Tumour Stem Cells

It has long been considered that glycolysis and mitochondrial oxidative phosphorylation independently promote tumourigenesis [102]. However, it is not difficult to see from the above description that many difficulties and doubts exist in exploring which metabolic phenotype participates in tumourigenesis. Therefore, research on these issues has provided us with more thought and inspiration to further understand the occurrence and development of tumours.

Glycolysis and OXPHOS may exert different effects on the initial stages of tumourigenesis and execute different functions. At the earliest stage of tumourigenesis, mitochondrial dysfunction may be an additional factor causing increased gene instability [103] and plays an important role in cell transformation. For example, poly [ADP-ribose] polymerase(PARP)-mediated poly-ADP ribosylation and sirtuin-mediated deacetylation can have important effects on genome stability and cell fate and are key factors that cause cell canceration [84]. However, it cannot be concluded that OXPHOS causes cell transformation at the early stage of tumourigenesis because the study also showed that the inhibition of OXPHOS does not affect the accumulation of tumour stem cells [83]. During tumourigenesis, functionally activated glycolysis is important for the cell’s energy supply, biosynthesis, and escape from death. Many studies have shown that glycolysis cannot only make up for the lack of energy supply caused by mitochondrial dysfunction, but also avoid ROS-induced apoptosis caused by OXPHOS during the initial stage of tumour development t [100,104,105]. Glycolysis can provide more substrate sources for promoting further development of tumour cells [106,107,108]. Therefore, abnormal changes, or the overactivation of OXPHOS, may be an important transformation process in tumorigenesis initiation [109,110,111] as the transformed cancer cells have strong growth demand, and the energy supply of glycolysis is far from sufficient. In contrast, OXPHOS can create the necessary conditions for maintaining the sustained growth and immortality of tumours by promoting aspartate synthesis, subsequent nucleotide metabolism, and other biochemical reactions [83,85,86]. OXPHOS is involved in the immortalization and further transformation of TIC, and reducing the level of OXPHOS can affect genes associated with the growth of cancerous tumours [83]. Furthermore, glycolysis may be required for cell differentiation, basic metabolism, and the preparation for proliferation or adaptation to the external environment [100,101,112] (Figure 1).

Both glycolysis and OXPHOS may be involved in the process of stem cell development into tumour stem cells, equipping tumour stem cells with the potential for metabolic heterogeneity. Many factors affect the survival and development of tumours, such as hypoxia, nutrition, and microenvironment [113,114,115]. Therefore, metabolic heterogeneity during tumourigenesis may be an adaptation to the environment [116], which may be one of the important reasons for detecting different tumour metabolic types. For example, regardless of whether the glioma microenvironment is close to the blood vessels that are supplied with sufficient oxygen area or the hypoxic regions at the distal end of blood vessels, all these conditions can accommodate CSCs [117,118,119], which indicates the presence of at least two metabolic phenotypes in CSCs [102,113]. Isogenic murine-induced single stem cells are built into GSCs (Ink4a/Arf–null, H-RasV12-expressing murine GICs), while CSCs are induced artificially to achieve the initial tumourigenesis conditions to observe the metabolic phenotype at the initiation to promotion stage. The results showed that there were two metabolic phenotypes: one is under homeostatic conditions; CSCs dependent on OXPHOS could switch their metabolism to glycolysis in oxygen availability, which confirmed the metabolic plasticity of a single tumour cell [120]. Another is that CSCs dependent on glycolysis have a higher affinity for hypoxic niches, but CSCs dependent on OXPHOS have a better reoxygenation affinity that shows a preference for perivascular regions, which indicates that heterogeneous tumour cells with different metabolic phenotypes in the tumour stem cell population exist [120]. Therefore, the metabolic heterogeneity exhibited by this plastic hybrid metabolic phenotype is an important metabolic regulatory basis for cancer cells to adapt to different microenvironments and meet their vigorous growth and proliferation [121,122,123,124] (Figure 1).

Third, the results may be unreliable if the complex interrelationship between glycolysis and OXPHOS is not fully or systematically clarified. On the one hand, even if the marker molecules of glycolysis or OXPHOS are significantly increased, it does not mean that their metabolic level is increased. For instance, some studies have reported that lactate dehydrogenase (LDH) is upregulated after cell cancerization. Following this concept, the upregulation of LDH may be more inclined to the increasing level of glycolysis, and related research and treatment will also be inclined to it [125,126]. However, later studies indicated that high levels of LDH did not cause the accumulation of lactic acid and extracellular acids [127]. They also found that LDH produced pyruvate from lactic acid to fuel the Tricarboxylic acid (TCA) cycle, while lactic acid originated from tumour cells driven by glycolysis [128,129,130]. The same results were also found that lactate secreted by Paneth cells can provide a carbon source for Lgr5^+^ ISCs to meet mitochondrial energy demands [131]. In addition, some researchers use isotopes to label some nutrients (for example, C13-labelled glucose) in some cells that use glycolysis as metabolic energy. It has been found that the intermediate products of the TCA cycle mainly come from labelled glucose (13C-glucose), pyruvate carboxylase (PC), and pyruvate dehydrogenase (PDH), which help to enhance the activity of the TCA cycle [132,133]. This is noteworthy because these tumour cells have the capacity for glycolysis as an energy source, indicating that the material source of the TCA cycle may be directly or indirectly derived from glycolytic intermediates and demonstrating that glycolytic tumour cells may realize their anabolic demand for biosynthesis by using the TCA cycle [132] (Figure 2). In addition, to study the occurrence and development of tumours from the metabolism perspective, focusing only on one point may not achieve the expected drug intervention effect. For example, the glycolytic level of glioma stem cells isolated from surgery is lower than that of differentiated progeny [134]. However, the inhibition of glycolysis with dichloroacetate could promote mitochondrial oxidative phosphorylation, which might result in an increasing effect on cell viability and number [135].

From the above description, we know that there is complex metabolic heterogeneity and regulation of metabolic plasticity in tumour cells. Metabolic heterogeneity indicates the presence of various metabolic phenotypes, whereas metabolic plasticity indicates that metabolic phenotypes can be mutually transformed. Therefore, a full understanding of the key approaches in the process of tumour metabolic transformation by mining the key molecules involved in the process of mutual transformation of glycolysis and OXPHOS may provide a clearer explanation of the above issues and provide better research means and more objective theoretical and experimental basis for the mechanism and treatment of tumour progression. Relevant studies have also been conducted. For example, a hybrid metabolic phenotype has been observed in triple-negative breast cancer (TNBC) cells. The authors developed a mathematical model to unearth node proteins that regulate the metabolic phenotype and found that AMPK and HIF-1α are core regulatory loop molecules involved in tumour metabolism. By analyzing the expression characteristics of these two proteins, they found that tumour cells display three stable metabolic states: W (HIF-1^high^/pAMPK^low^), O (HIF-1^low^/pAMPK^high^), and W/O (HIF-1^high^/pAMPK^high^), corresponding to the glycolytic phenotype, OXPHOS phenotype, and hybrid phenotype [136], which enables tumour cells to switch between OXPHOS and glycolytic phenotypes. By inhibiting the expression of AMPK and HIF-1, the ability of metabolic plasticity of cancer cells can be effectively eliminated. In this model, the team extensively studied the diversity of tumour metabolism and realized a more effective killing of tumours by mining node molecules that regulate metabolism and the use of targeted drugs, which may become a practical and reliable way to solve the clinical drug resistance of tumours.

## 3. Molecules Affecting Tumour Glycolysis and OXPHOS Transformation and Significance

### 3.1. Effect of Glycolytic Rate-Limiting Enzyme on Metabolic Plasticity

Glycolysis is mainly regulated by rate-limiting enzymes that catalyze three irreversible reactions: hexokinase (of the four HK isoenzymes, HK-2 has the most significant correlation with tumour cells) [137,138,139,140], phosphofructokinase (PFK), and pyruvate kinase (PK). Once glucose enters the cell, glycolysis is catalyzed by HK, and PK catalyzes the “export”. PFK is the most important rate-limiting enzyme of glycolysis, which can quickly react with ATP/ADP, citric acid, and other alternative fuels, such as fatty acids, ketones, and hormones, to regulate the whole process of glycolysis [141].

#### 3.1.1. Hexokinase

Several in-depth reports on the effects of the three rate-limiting enzymes on metabolic plasticity are available. Early studies found that in some non-cancer cells that were dominated by glycolysis, the deletion of HK-2 would inhibit the glycolysis pathway, leading to ^13^C-glucose accumulation in the cytoplasm. Then, the number and size of mitochondria expanded, the TCA metabolic pathway was enhanced [142,143], and the activity of mitochondrial electron transport chain also increased [144,145].

However, this does not imply that the cell alleviates the metabolic pressure caused by the lack of HK-2 through the above metabolic pathway. HK-2 is not only the first rate-limiting enzyme of glycolysis, but it can also bind to the voltage-dependent anion channel (VDAC) on the outer membrane of the mitochondria. Mitochondrial VDAC is a channel for the binding and release of many apoptotic molecules, such as Cytochrome c and BCL2-associatedx(Bax) [146,147,148]. The combination of HK-2 and VDAC can, utilizing ATP produced by still-functioning mitochondria, realize the role of self-phosphorylation activation [149]. However, combining these two proteins can prevent some pro-apoptotic molecules from binding to VDAC, thus avoiding the initiation of mitochondrial apoptosis [150]. Therefore, the regulation of the metabolic plasticity process from glycolysis to OXPHOS, caused by the deletion of HK-2, is accompanied by the production of more ROS, the release of apoptosis-promoting molecules, and the accumulation of metabolites in the TCA cycle, which are also signs of mitochondrial dysfunction and apoptosis induction [151,152,153]. Based on this, some scientific researchers have developed precise combinations of drugs with HK-2 inhibitors, which have produced more effective killing effects on tumour cells. In general, the conversion of glycolysis to OXPHOS, caused by HK-2 inhibition, can be targeted by tumour drugs in two ways. One example is increasing tumours’ sensitivity to apoptosis during this process. Zhang et al. found that the selective inhibition of HK-2 expression by follicle-stimulating hormone (FSH) peptide-aligned PEG PEI copolymers loaded with HK2 shRNA could switch the metabolism from glycolysis to OXPHOS. In contrast, the expression of pro-apoptotic molecules increased. Therefore, treatment with chemotherapy drugs, such as cisplatin, can improve the killing effect of anticancer drugs on ovarian cancer [154]. Another example is changing the drug resistance of tumours. Dannielle et al. reported that the loss of HK2 could lead to an increase in OXPHOS, and the use of metformin in this process could simultaneously inhibit the activity of Complex I in OXPHOS, which can considerably reduce the drug resistance of tumour cells to sorafenib [155]. This observation highlights the impact of HK on metabolic remodelling and potential treatment options.

#### 3.1.2. Phosphofructokinase

In contrast to HK, which is the most important rate-limiting enzyme of glycolysis, targeted drugs acting solely on PFK may have the same anti-cancer effect [156]. For instance, mitochondrial VDAC can bind not only with HK-2 but also with PFK [157,158]. The combination of VDAC and PFK has a significant inhibitory effect on PFK-mediated glycolysis [157], thus affecting the metabolic phenotype of tumour cells.

As previously reported, the metabolic phenotypes of GSCs and non-stem tumour cells (NSTCs) are very different. GSCs are more inclined to glycolysis than NSTCs, and their mechanism is realized through VDAC. NSTCs express higher levels of VDAC, which can be combined with PFKP, a dominant isoform of PFK in human glioblastoma (GBMs), making it bind to the outer membrane of mitochondria, thus inhibiting the expression of phosphofructokinase, platelet (PFKP) in the cytoplasm and inhibiting its glycolysis reaction [157,159]. Once VDAC is lost, PFKP is released, which activates glycolysis and causes the conversion of the metabolic phenotype [157]. Using this mechanism, clotrimazole, a PFK inhibitor, can alleviate the activation of glycolysis caused by VDAC disruption, thereby preventing the metabolic rearrangement process and phenotypic transformation of GSCs, as well as significantly inhibiting the growth and invasion of glioma cells [157]. This treatment mode has transformation significance because GSCs have strong drug resistance to current radio and chemotherapeutic regimens due to their strong cell plasticity regulation, which limits the treatment of many antineoplastic drugs.

In addition, avoiding apoptosis by regulating metabolism is a feature of tumour survival [160]. The Warburg effect is the cornerstone for tumours to realize the above process [161]. Therefore, by targeting the regulation of PFK expression to inhibit glycolysis and to improve the level of OXPHOS in a variety of tumour cells (such as pancreatic ductal adenocarcinoma (PDAC), non-small cell lung cancer (NSCLC) cells, and CRCs), improve the release of ROS and apoptosis-promoting molecules, and, finally, enhance the sensitivity of anti-cancer drugs, good anti-tumour effects have been achieved [158,162,163]. For example, DNASE1L3 (also named DNase1) is a DNA-degrading enzyme expressed at low levels in high-level glycolytic tumour cells [164]. A high expression of DNASE1L3 can induce tumour apoptosis [165], which is considered a prospective and personalized gene-based therapy for cancer [166,167,168]. The addition of DNASE1L3 can inhibit PFK1-mediated glycolysis and induce tumour apoptosis by activating the P53 pathway, thus inhibiting tumour growth, invasion, and migration [168]. Based on the above mechanism, human recombinant DNase1 (hrDNase1) can achieve high killing efficiency in tumour cells resistant to radiotherapy and chemotherapy [169]. Another example is that TNF receptor-associated Protein 1(TRAP1), a heat shock Protein 90 (HSP90) molecular chaperone that is upregulated in human CRCs and responsible for the downregulation of OXPHOS [163,170], can combine with PFK1, avoid the ubiquitination of PFK1, and enhance glycolysis activity, thus producing resistance to some drugs against glycolysis, such as cetuximab [163]. The inhibition of TRAP1 reduces its binding with PFK1, triggering the degradation of PFK1 through the ubiquitin-proteasome pathway. The downregulation of TRAP1 expression induces plasticity regulation, ultimately enhancing the sensitivity of CRCs to cetuximab [163].

#### 3.1.3. Pyruvate Kinase

Pyruvate kinases have several isoforms. Compared with other pyruvate kinases, the M2 isoform of pyruvate kinase (PKM2), which is a specific PK of tumour cells, is significantly upregulated in tumours [171,172].

Compared to the above two rate-limiting enzymes, the effect of PKM2 on metabolic remodeling is more complex. The presence of PKM2 in tumour cells can inhibit OXPHOS by inhibiting the expression of monoubiquitinated histone H2B (H2Bub1) and enhance pyruvate phosphorylation in glycolysis to influence metabolic rearrangement. H2Bub1 can regulate a series of mitochondrial genes used for OXPHOS, thereby affecting the production of mitochondrial ATP and inhibiting glycolysis in human lung cancer cells (H1299 and A549 cell lines). PKM2 can combine with it to inhibit the expression level of H2Bub1 [173]. However, tumour cells can reduce mitochondrial fragmentation, promote mitochondrial fusion, protect mitochondrial function, and enhance OXPHOS by promoting PKM2 expression [174,175].

Therefore, PKM2 may have different beneficial effects on glycolysis and mitochondrial oxidation. The different functions of PKM2 may not depend on its specific protein activity patterns. PKM2 forms different polymers, including dimers and tetramers. Tetramers are the main functional and active units of PKM2 that play a role in glycolysis, providing energy for tumour growth and proliferation. Inactivated PKM2 is a dimer that can be oxidized and phosphorylated, resulting in the loss of catalytic function during glycolysis. However, it can enter the nucleus and combine with other transcription factors to exert transcriptional activity [176,177] and participate in the transcription and regulation of many molecules, affecting tumour proliferation, migration, and death [178,179]. For example, Dimer PKM2 can promote the release of exosomes through phosphorylating synaptosome-associated Protein 23 and affect the signal transduction between tumour cells and cells, and the microenvironment [180]. By inhibiting the encoding of E-cadherin and increasing the nuclear transcription of signal transducer and activator of Transcription 3 (STAT3), Dimer PKM2 can lead to a loss of EMT in tumour cells and promote tumour cell invasion and migration [181,182]. In addition, Dimer PKM2 can also mediate the disruption of NF-κB/miR-148a/152 and promote the expression of vascular endothelial growth factor (VEGF) as well as the activation of insulin like Growth Factor 1 receptor (IGF-IR), thereby promoting tumour angiogenesis [183,184,185,186].

Therefore, when PKM2 is used for drug intervention, more targeted measures are required to achieve better results. The oxidation and phosphorylation of PKM2 result in its nuclear localization and interaction with HIF-1α, which causes EMT-related molecules’ expression and increases tumour cell invasion [187]. Under these conditions, tumour cells are more likely to undergo OXPHOS reactions through lactic acid produced by cancer-associated fibroblasts (CAFs) dominated by glycolysis in the tumour microenvironment [187,188]. Using DASA, which can reactivate PKM2, could avoid its nuclear localization and enable tumour cells to restore glycolysis and reduce its function in EMT transcription. However, tumour cells can still maintain sufficient OXHPOS levels. When tumour cells are cultured in a non-sugar environment and treated with metformin, an inhibitor of mitochondrial complex I(COXI) activity, the growth and proliferation of tumour cells can be significantly inhibited [187]. Additionally, researchers have found that some tumour cells with high levels of both OXPHOS and glycolysis conduct multiple drug interventions based on metabolic phenotypes [175]. For example, the use of MLN4924, a small-molecule inhibitor of neoplasia modification, could induce the conversion of mitochondrial fission to fusion, thereby inhibiting the TCA cycle. Shikonin can inhibit the tetramer PKM2-mediated glycolysis reaction. This drug combination has achieved good results in both in vivo and in vitro experiments [175]. Due to its anti-cancer effect, MLN4924 is used as a clinical medication in Phase I/II [189]. Based on the mode of regulation of glycolysis-related molecules at the entry point, changing the metabolic phenotype of tumour cells and then using drug intervention according to the molecular regulation mechanism can be used to develop precision therapy.

### 3.2. Effects of Key Enzymes of Mitochondrial Oxidative Metabolism on Metabolic Plasticity

Mitochondrial oxidative phosphorylation is the ultimate destination for oxygen molecules during aerobic metabolism [190]. Energy substances are metabolized through the TCA cycle, which is the key link that causes a significant increase in OXPHOS levels [191,192,193]. Abnormal changes in key enzyme molecules, such as mutations, abnormal expression, and activity damage, lead to the accumulation of Krebs cycle intermediates [194,195], thus affecting the vigorous and diversified growth needs of tumour cells [196,197]. At the same time, the metabolic process has an important impact on the key molecules and events of metabolic transformation, such as HIF-1, AMPK, and ROS, to promote the metabolic transformation between glycolysis and OXPHOS [198,199,200,201]. Owing to space limitations, this article lists three key enzymes of the TCA cycle, namely citrate synthase (CS), isocitrate dehydrogenase (IDH), and α-ketoglutarate dehydrogenase complex(α-KGDC), to describe the influence of related enzymes on the metabolic plasticity process and the status of some targeted drug therapies.

#### 3.2.1. Key Enzymes of the TCA Cycle

##### Citrate Synthase (CS)

CS can convert Acetyl Coenzyme A and oxaloacetic acid to citric acid. Citric acid is an important intermediate that affects metabolic transformation and can produce an important allosteric inhibition effect on PFK1/2 [202,203], thereby reducing glycolysis [204]. The insufficient energy supply caused by the inhibition of glycolysis due to the excessive increase in citric acid can be attributed to mitochondrial oxidative metabolism. In mitochondria, citric acid is a key regulator of mitochondrial oxidation enzymes, such as successful dehydrogenase (SDH), PDH, and Carnitine Palmitoyl Transfer I (CPT) [203,204,205,206,207]. However, as previously mentioned, the increase in the TCA cycle and the enhancement of OXPHOS caused by metabolic remodeling may also be features of mitochondrial dysfunction in tumour cells, and citric acid plays an important role in this process [151,152,153]. Citric acid can cause a decrease in the expression of anti-apoptotic molecules, such as BCL-2 and myeloid cell leukaemia 1(Mcl-1) [208,209,210], and an increase in the expression of apoptosis-initiating and executing molecules, such as caspase family-related proteins (caspase-2/3/8/9), which finally inhibit tumour cells [211,212].

Therefore, the inhibition of different types of tumour cells by regulating citrate synthase and its catalytecitric acid has been reported in many studies. For instance, oral citric acid has a good inhibitory effect on lung cancer [213,214], thyroid cancer [215,216], and pancreatic cancer [217]. Citric acid has a synergistic anti-tumour effect with some drugs, such as 3-bromo-pyruvate [212], cisplatin [210], and celecoxib [218], which can effectively improve the lethality of anti-cancer drugs. The effectiveness of anti-cancer drugs can also be greatly improved if metabolic plasticity is considered. For example, when T98G glioma cells were cultured in a medium containing galactose and an increased concentration of glucose (DMEM GAL medium), the citrate synthase activity was increased by 26%, and the corresponding citric acid was also increased. Correspondingly, TCA increased, and the level of glycolysis was weakened, reflecting the metabolic remodeling process from glycolysis to OXPHOS [205]. The authors believe mitochondria may become more sensitive to some mitochondria-targeted drugs under this condition. Therefore, the MI–D compound [4-phenyl-5-(4-nitro-cinnamoyl)-1,3,4-thiadiazolium-2-phenylamine], a potential candidate mitochondria-targeting drug [219], was used. They found that the MI–D compound can effectively enhance the activity of citrate synthase and citric acid and induce tumour metabolic remodelling, and, correspondingly, the tumour-killing effect of the drug is more obvious [205]. The above results explain the mechanism by which MI-D, an anticancer drug, can produce a more effective tumour-killing effect under the condition of inducing tumour metabolic remodelling. The above results reflect the effective antitumour effects of anti-tumour drugs under metabolic remodeling.

##### Isocitrate Dehydrogenase (IDH)

The effect of IDH on tumour cells mainly depends on the mutation state. Early reports stated that more than 70% of IDH mutations have been found in the middle- and high-grade (Grade II–III) astrocytomas, oligodendrogliomas, and secondary glioblastomas [220,221,222,223]. Later, IDH mutations were found in the early stages of some tumour cells, such as acute myeloid leukaemia and solid tumours [224,225,226,227]. Based on its important diagnostic value after mutation, it was defined as the classification standard for central nervous system tumours by the WHO in 2016 [228]. In the latest classification standard of central nervous system (CNS) tumours by the WHO in 2021, invasive glioma was clinically classified according to IDH1/2 mutations [229]. Therefore, to explore the effect of IDHs on tumour metabolic remodeling, we should focus on the effect of IDH mutations. Mammals mainly express IDH1, IDH2, and IDH3. IDH1 and IDH2 have more than 70% homology, and a recent report found that IDH1/2 is closely related to tumours. Wild-type IDHs (WT-IDHs) can catalyze the conversion of isocitrate to α-ketoglutarate (α-KG), CO_2_, and a certain amount of NADPH. The mutant IDH, especially mutation at the R132H site, can greatly change the enzyme activity, enhance its ability to bind to NAPDH, and reduce its affinity for isocitrate, which ultimately leads to its inability to catalyze isocitrate to α-KG and make the (R) enantiomer of 2-hydroxyglutarate(R-2HG) generate and release [230,231,232,233].

R-2HG is an oncometabolite [234,235,236,237] and is one of the best small molecule biomarkers for tumours [238,239]. R-2HG can cause abnormal expression of many molecules related to metabolic enzymes. For example, many studies have reported that R-2HG increases HIF-1α expression through its inhibitory effect on PDH, then enhances the HIF-1α-mediated glycolytic pathway [240,241]. However, it has also been reported that R-2HG can activate PDHs and degrade HIF-1α [242]. In addition, it has been reported that R-2HG can inhibit aerobic glycolysis by abolishing the FTO/m6A/YTHDF2-mediated post-transcriptional modification of PFKP and Lactate Dehydrogenase B (LDHB) [243]. In addition, R-2HG can also make mitochondria more susceptible to glucose starvation by inhibiting ATP production through its inhibitory effect on the oxidative phosphorylation of mitochondria [244].

However, the effect of IDH1/2 mutations on metabolic remodeling is not achieved only by inducing R-2HG. Many reports have shown that IDH1/2 mutation can increase the level of lactate, such as lactate levels in ^mut^IDH1 (R132H) in U87 GBM [245], ^mut^IDH1 (R132H), and ^mut^IDH2 (R172K) in Grade II and III glioma [246]. Other reports have shown its inhibitory effect on lactate, such as ^mut^IDH1 (R132H) in U87, NHA, and LN18 cells [247,248,249], and not changed in many other reports, such as ^mut^IDH2 (R172K) in HOG cells [250]. Pyruvic acid has also been reported differently in relevant studies, such as unchanged studies on ^mut^IDH1 (R132H) glioma and PDX mouse models, as well as in ^mut^IDH1(R132H) expressing LN18 or HOG cell lines [249,250,251,252], and decreased studies, such as in ^mut^IDH1 (R132H) PTBs [252,253]. In the TCA cycle, the mutation of IDH1/2 can also cause inconsistent changes in some metabolic intermediates, such as 2-OG, citrate, cis-aconite, isocitrate, and fumarate [245,249,250,251,252,253,254]. However, compared with WT–IDH cells that are dominated by glycolysis [255,256], IDH mutant tumour cells have a stronger TCA cycle [256,257,258], more OXPHOS [259], and lower glycolysis levels [256,260,261]. This may be related to using different carbon sources after the IDH mutation. Tumour cells with IDH mutation did not use glucose for oxidation [248,256,262,263]. Instead, they used lactic acid and glutamate as substrates for TCA metabolism [248,256,264] and metabolized pyruvate by increasing the expression of PC, which increased the production of oxaloacetate and succinate [258,265], thereby meeting the needs of the electronic transfer chain.

The different lactate levels may also be related to the uptake and release of lactic acid in IDH mutation tumour cells. Tumour cells with IDH mutation will express less MCT1/4, LDHA also decreases, and, thus, the production of lactic acid is reduced. However, pyruvate cannot be obtained from glycolysis due to the IDH mutation, which needs to be obtained from the tumour microenvironment or intracellular metabolism of lactic acid to provide a carbon source for TCA. Researchers have proposed that to study IDH1/2 mutant tumour cells, it is necessary to observe the dynamic intracellular and extracellular lactic acid content, but the current detection methods are not well-implemented [248,251,252].

Therefore, for IDH-mutated tumour cells, more intelligent treatment measures may be needed to achieve better drug intervention effects. For example, studies have shown that for acute myeloid leukaemia (AML) patients with IDH1 mutation, the use of drugs targeting the inhibition of mutant IDH1 alone may cause patients to be insensitive to drugs, mainly due to the inhibition of mutant IDH1, which could also cause the improvement of glycolysis [266,267]. Therefore, some researchers have identified several characteristic genes related to glycolysis by constructing IDH1-mutant GBM cells. Then, the risk scores of these genes were integrated and analyzed to correlate them with chemotherapy drugs, such as cells with mutant IDH1 having better sensitivity to doxorubicin and sorafenib [268]. A combination of glycolysis and mutant IDH inhibitors has been applied to treat other tumours to achieve better therapeutic efficacy [243,259].

However, the treatment mode cannot be realized simply by using multiple drugs, and certain conditions may need to be created before using a combined drug. For example, some tumour cells, such as IDH1-mutated CSCs, are mainly metabolized by OXPHOS. However, because they are in a long-term hypoxic state, drugs targeting OXPHOS are ineffective and retain glycolysis ability simultaneously. Therefore, the author proposes removing cells from the hypoxic area and then using the corresponding radiotherapy and chemotherapy methods, which can produce more ROS. This treatment has achieved good results [269,270]. In addition, other researchers have developed new drugs, such as BAY 87-2243 (B87), a special inhibitor of HIF-1α [271], which can also inhibit the activity of Complex I [272]. The oxygen consumption rate (OCR) level of cells decreases after B87 treatment, which reflects the inhibition of OXPHOS [273,274]. Although the drug can inhibit the proliferation of tumour cells, B87 is not very effective in inducing tumour cell death [274]. Later studies found that tumour cells treated with B87 had obvious anti-tumour effects under the environment of reduced glucose or combined with some glycolysis inhibitors, such as 2-deoxyglycose, indicating that the cancer cells treated with B87 were saccharophilous [20,275]. Inspired by this, researchers used synergistic molecules/drugs for B87 drugs and found that the α-KG precursor dimethyl-α-ketoglutarate (DMKG) can produce a strong tumour-killing effect together with B87 [274].

In addition to the above examples, compared with mutant IDH tumour cells, wild-type IDH has higher levels of glycolysis and TCA [255,276], which mainly depends on glucose and glutamine metabolism to generate pyruvate and α-KG [248,277,278]. Moreover, the survival rate of patients with wild-type IDH tumours is not as good as that of IDH mutant patients [279,280], which may be related to the different responses of these tumour patients to IDH inhibitors [267]. One of the more important mechanisms is that IDH can produce antioxidants and promote oxygen-sensitive signal transduction pathways [281,282,283]. However, for mutant IDH tumour cells, due to a lack of antioxidants and a high level of the TCA cycle, these tumour cells could produce more ROS [284,285], which leads to an increase in the sensitivity of radiotherapy and chemotherapy to mutant IDH patients, but not to the susceptibility of wild-type IDH tumour cells [270,286]. Therefore, according to the metabolic characteristics of IDH wild-type tumour cells, using drugs that induce ROS can produce better anti-cancer effects [286].

##### α-Ketoglutarate Dehydrogenase Complex(α-KGDC)

α-KGDC is a key regulatory point in the tricarboxylic acid cycle and is mainly used to catalyze α-KG, succinyl coenzyme A, and CO_2_, as well as convert NAD^+^ to NADH, which can be used later in the respiratory chain complex to produce ATP. Although many factors, such as pH, calcium ion concentration, and ROS [259], can regulate the activity of α-KGDC, current research on the direct effect of α-KGDC on metabolic plasticity is rare. The existing reports showed that the most important effect of α-KGDC is the mutual feedback loop regulation mode between its metabolic substrate α-KG and HIF-1. Under normal oxygen conditions, α-KG produced by α-KGDC can activate PHDs and hydroxylate HIF-1α, while hydroxylated HIF-1α can be degraded by the proteasome pathway [287,288,289,290]. However, in the absence of oxygen, the production of α-KG is limited, thus inhibiting the activity of PHDs. The degradation of HIF-1α decreases and its accumulation increases, which then targets the regulation of the E1 subunit of α-KGDC so that the E1 subunit is ubiquitinated under the action of ubiquitin-protein ligase SIAH2, resulting in its degradation [288,291]. The inhibition of α-KGDC leads to α-KG failure to persist in the mitochondrial TCA cycle. Instead, it enters the reductive pathway of the TCA cycle to produce acid, which ultimately promotes lipid synthesis. HIF-1α is an important promoter of glycolysis, which has a positive transcriptional regulatory effect on various glycolytic molecules, such as HK2, GLUT1, and PDK1 [292,293,294]. Therefore, the HIF-1α mediated regulation of α-KG function under hypoxia may have a potential anti-tumour effect [295,296].

In addition, the inhibition of α-KGDC activity leads to the accumulation of α-KG, which also affects the regulation of autophagy. Report presents that a decrease in mitochondrial oxidative phosphorylation is accompanied by a low level of α-KG, while a high level of α-KG can inhibit autophagy through its target regulation of rapamycin 1 signaling [297,298]. Autophagy can affect the process of metabolic plasticity in many aspects. For instance, the polyubiquitination of HK-2, catalyzed by E3 ligase TRAF6, can be recognized by autophagy Protein P62 before initiating the degradation enzyme of the autophagy pathway, thereby inhibiting glycolysis [299]. Mitochondrial autophagy can reduce glycolysis, increase OXPHOS, increase cell apoptosis, and inhibit ectopic metastatic growth of cancer in a P53-dependent manner [300]. Moreover, some studies have shown that the significance of autophagy induction lies in the fact that tumour cells can alleviate survival disorders caused by metabolic and therapeutic stress [301,302,303,304,305]. However, regardless of how autophagy affects tumour cells, in the selection of tumour-targeting drugs affected by α-KG, we can consider whether or how the activation process of autophagy is by α-KG. Then, we can use some small molecule drugs to target autophagy [306] and combine them with drugs to α-KG, such as JHU-083 [276], which may have a better curative effect.

### 3.3. Other Molecules Regulating Metabolism Plasticity Processes

In addition to the effects of key metabolic enzymes on metabolic plasticity, some molecules of non-rate-limiting enzymes also have an important impact on the metabolic plasticity process, thus providing potential value for tumour treatment (Figure 3). For example, FBP2, the isoenzyme of FBP (fructose-1, 6-bisphosphatase), is a molecule that can regulate the rate-limiting enzyme of glycolysis and inhibit the proliferation of soft tissue sarcomas. There are two mechanisms involved in its inhibition process. A high expression of FBP can inhibit HIF-1α, interfere with glycolysis, and hinder tumour cell proliferation, invasion, and migration [307]. While FBP2 can also be localized in the nucleus, with c-Myc, it exerts an inhibitory effect by binding to the promoter of mitochondrial transcription factor nuclear respiratory factors (NRFs) and TFAM, thereby limiting the expression of mitochondrial genes and reducing OXPHOS energy supply. This dual inhibition function provides a good direction for tumour treatment [307,308].

Another example is the influence of some non-key enzymes of the TCA cycle and mitochondrial proteins on metabolic plasticity. For instance, mutations in SDH in hereditary paragangliomas and fumarate hydratase (FH) in leiomyomatosis and kidney cancers lead to the accumulation of succinate and fumarate [309,310], thus reducing the degradation of HIF and promoting the expression of multiple glycolytic enzyme genes [311]. In renal cell carcinoma, the inhibition of HSP60, a chaperone protein that plays an important role in mitochondrial protein homeostasis, can strengthen the glycolysis reaction, reduce OXPHOS and the transition from glycolysis to OXPHOS, and enhance de novo nucleotide synthesis and lipid synthesis. Therefore, high HSP60 expression is related to a better prognosis of renal cell carcinoma, which can lead to the inhibition of cancer cell growth and proliferation [312].

In addition, regulating related molecules in the electron transport chain pathway is also one of the ways to change tumour metabolism. By directly regulating decaprenyl diphosphate synthase subunit 2 (PDSS2), a mitochondrial metabolic molecule that plays a crucial role in regulating the synthesis of coenzyme Q-binding protein 10 (CoQ10), the activity of CoQ10 and mitochondrial Complex I can be increased, and the TCA cycle activity can be enhanced, thus inducing the conversion of glycolysis to mitochondrial respiration in hepatoma cells. This effect can inhibit foci formation, colony formation in soft agar, and tumour formation in nude mice [313].

In addition, the stability of genes, such as tumour protein p53 (TP53), KRAS, kelch-like ECH-associated protein 1 (Keap1), serine/threonine kinase 11 (STK11), EGFR, neurofibromin 1 (NF1), and B-Raf proto-oncogene, serine/threonine kinase (BRAF) is related to metabolic changes [314,315,316], the most important of which is the effect of the p53 gene on metabolism. At present, the effects of P53 on metabolic remodeling have been studied in many aspects, such as TP53-induced glycolysis regulatory phosphatase (TIGAR), glycolysis, and glucose transporter; the synthesis of cytochrome c oxidase 2 (SCO2) and glutaminase 2 (GLS2); or protein interactions with metabolic enzymes, such as glucose-6-phosphate dehydrogenase (G-6-PD), peroxisome proliferative-activated receptor, gamma, coactivator 1 (PPARGC-1), and sterol regulatory element-binding protein (SREBP) [317,318]. Although for tumour cells, the effect of TP53 on metabolism plasticity has different aspects, many studies have shown that TP53 can inhibit tumours by inhibiting glycolysis and promoting OXPHOS [319]. Another study reported that TP53 could suppress tumorigenesis by inhibiting the conversion of OXPHOS to glycolytic processes during tumour initiation [318]. We believe that wild-type P53 (WTP53) is an anti-oncogene. The function of P53 in preventing the cell from canceration is related to its ability to inhibit OXPHOS. This regulation pattern of WTP53 could reduce harmful products, such as ROS, and obtain more stable genomic DNA, thus avoiding the initiation of tumorigenesis. It has a similar role on normal somatic cells that rely on OXPHOS [320,321,322,323]. However, due to the existence of a mutation in TP53 (MuTP53), the positive regulation of OXPHOS could be lost, causing the occurrence of tumours (Figure 4). The beneficial effect of WTP53 on cell survival has also been preserved in tumour cells, although more than 50% of human cancers have mutations in WTP53 [324]. In hepatocarcinoma (HCC), the mutation frequency of WTP53 is low [325]. By introducing WTP53, the mitochondrial pyruvate carrier (MPC) binding to pyruvate can be disrupted, which is realized by inducing BCL2 binding component 3 (Bbc3,also named PUMA) and inhibiting the uptake of pyruvate by mitochondria, thus promoting the survival of tumour cells and causing poor prognosis in HCC patients [326]. Therefore, these findings suggest that treatment or research measures for cancer should consider activating WTP53.

### 3.4. Summary

Studying the enzymes and related molecules directly involved in regulating glycolysis and mitochondrial oxidation may provide a more accurate perspective on regulating metabolic plasticity and developing effective therapeutic measures. The above section lists some promising reports that can be potentially used as drug therapies targeting metabolic plasticity (partial drugs are summarized in Table 1). Due to limited content, this article only lists the essential enzymes and introduces some important molecules (partial molecules are summarized in Table 2).

## 4. The Role of Metabolic Plasticity Regulation in the Tumour Microenvironment

Genes that regulate metabolic plasticity in tumour cells are the basis of metabolic remodeling in tumourigenesis. The tumour microenvironment is an important factor affecting metabolic plasticity, tumorigenesis, and development. In the advanced stage, the tumour volume at the primary site develops to a certain extent, and a microenvironment conducive to its survival has been formed. With further expansion of the tumour volume, the space and nutrition source of the primary site cannot meet the rapid growth of the tumour, and it begins to prepare for the next step, that is, diffusion; the process from promotion to progression is a stage in which tumour cells fight against immune cells and compete for nutrients in the surrounding environment to remodel and create the surrounding microenvironment (Figure 5).

### 4.1. Effects of Metabolic Plasticity Regulation on the Immune Microenvironment

Inducing the differentiation of naive CD8^+^ T cells is the primary anti-tumour mechanism of immune cells. Naive CD8^+^ T cells normally exist under quiescent conditions. Maintenance of this state is mainly mediated by two molecules: sphingosine 1-phosphate (S1P) and interleukin (IL)−7 [327,328]. S1P is important for OXPHOS [328,329,330], while IL-7 mainly promotes glucose uptake by GLUT-1, thus affecting the glycolytic process [331,332,333]. Glycolysis is primarily used for the activation of T cells, but OXPHOS is also important, and its deletion blocks the proliferation of T cells [334,335]. Once the anti-cancer signal is received, naive CD8^+^ T cells differentiate and become effector CD8^+^ T cells and memory CD8^+^ T cells. Compared with effector CD8^+^ T cells, memory CD8^+^ T cells have better mitochondrial activity [336]. Upon T cell receptor (TCR) and leukocyte surface differentiation antigen 28 (CD28) stimulation, acylglycerol kinase (AGK) can be combined with phosphatase and tensin homolog (PTEN), which leads to phosphorylation of the lipid phosphatase PTEN and loss of its activity, thereby activating the phosphatidylinositol 3-kinase (PI3K) mechanistic target of rapamycin kinase (mTOR)-signaling pathway, promoting its glycolytic metabolism, and differentiating into effector T cells [337], exerting its anti-tumour effect.

Overall, immune cells in the tumour microenvironment, and tumour cells, have similar metabolic modes. The two compete for resources to inhibit the other’s survival. At the initial stage, when resources are sufficient, T cells use glucose for glycolysis and mitochondrial metabolism to generate T cells with tumour-killing effects, which involves the activation of HIF-1α and/or Myc [10]. Cancer-associated antigens (CAAs) produced by the death of tumour cells can be taken up by dendritic cells (DCs) and transmitted to naive T cells in lymph nodes with tumour infiltration. After stimulation by CAAs, differentiated effector CD8^+^ T cells move and infiltrate the tumour area, recognize the tumour through the relevant antigens on the surface of tumour cells, and initiate the death of the tumour.

The dead tumour cells release CAAs and repeat the above process [338], called the cancer immunity cycle (CIC). Failure of the CIC process leads to the immune escape of tumour cells [20]. CIC realizes the above process through two points: immunogenicity of the tumour itself and the effector function of infiltrating T cells. Sufficient tumour mutations can increase immunogenicity. The uptake of glucose, glutamine, and other nutrients can increase the mutation effect [339,340]. In addition to the uptake of nutrients, which can be utilized by glycolysis, OXPHOS can also be used for its metabolism. At the same time, due to the specificity of mitochondrial function, the process of mitochondrial oxidation can produce ROS, causing more damage to tumour cells and inducing DNA mutations [341,342,343,344,345]. PI3K, Myc, RAS, and programmed cell death 1 ligand 1 (PD-L1) are the main pathway molecules involved in this mutation process [345,346,347]. In addition, aerobic glycolysis may have an inhibitory effect on tumour immunogenicity [348]. The transition of human OXPHOSS to glycolysis can reduce the occurrence of mutations caused by ROS (nuclear and mitochondrial) [349], lactic acid produced by glycolysis can reduce the immunogenicity of cells [37,350,351,352], and increased glycolysis can inhibit the synthesis of antigenic peptides by increasing cell cycle progression [353,354]. Therefore, the change in metabolic mode caused by various factors is an important inducer of increased tumour immunogenicity.

In addition, with the growth of a tumour and the intake of more glucose, the expression of high levels of Myc and HIF can increase the glycolysis flux of tumour cells, resulting in the excessive release of immunosuppressive molecules such as lactic acid and inhibitory cytokines in the TME, which has a significant impact on the activation of effector T and Treg cells. Lactic acid is an inhibitory molecule for all anti-tumour immune cells. Effector T cells in humans and mice lost their killing ability to tumour cells in the lactate microenvironment [35,355,356]. However, the high lactate environment can enable Treg cells to absorb lactate through monocarboxylate transporter 1 (MCT1) and enhance the expression of PD-1, thereby activating regulatory T (Treg) cells [34,357]. An increase in activated Treg cells can inhibit effector CD8^+^ T cells [358,359]. Therefore, when MCT1 on the surface of Treg cells is deleted, lactate importation can be disrupted, and anti-tumour ability can be improved [34].

The tumour microenvironment is a complex cellular environment. Besides lymphocytes, macrophages, CAFs, and natural killer (NK) cells are also important components [360,361]. Macrophages play a role in all stages of tumourigenesis and development [362]. A high level of lactic acid can make macrophages develop into M2-type macrophages, inhibit T cell activation and proliferation, and exert its immunosuppressive function by expressing arginase 1 (ARG1) protein [363,364,365]. Macrophages intaking glucose can develop into M1-type macrophages after receiving interferon gamma (IFN-γ) secreted by Th1 cells [364], and the anti-tumour ability of M1-macrophages can be enhanced. However, IFN-γ is unstable in an acidic microenvironment, thus inhibiting the differentiation of Th1 cells to Th2, which antagonizes the differentiation of M1-type macrophages and promotes tumorigenesis [366,367,368,369]. Macrophages affected by lactic acid can also secrete IL-6, increasing 3-phosphoinositide dependent protein kinase 1 (PDPK1)-dependent phosphoglycerate kinase (PGK1) phosphorylation by inducing PDPK1 in tumour cells, resulting in enhanced glycolysis and tumourigenesis [370]. Therefore, it is not difficult to see that the deprivation of surrounding nutrients and elimination of metabolites by the tumour affect the antitumour effect of macrophages from a long-term perspective.

Similar results were also found for NK cells. At the initial stage of KRAS-driven lung cancer, NK cells can exhibit strong cytotoxic effects and secrete cytotoxic factors to kill target cells directly [371]. This process is closely related to metabolic changes. That is, the activation of NK cells needs to switch their metabolism from mitochondrial oxidation to glycolysis [372]. However, for advanced tumour cells, the function of NK cells in tumour cells infiltrated in tumour cells, but not in blood circulation, is suppressed, reflecting the regulatory effect of the tumour microenvironment on NK cells. Mechanistically, FBP1 weakens the glycolytic levels of NK cells, thereby causing dysfunction [373,374]. While FBP1 is mediated by transforming growth factor beta (TGF-β) in the tumour microenvironment [375], CAFs are the origin of TGF-β, and CAFs are the most abundant cells in the tumour stroma that are recruited by platelet-derived growth factor secreted through the tumour [376]. The emergence of CAFs generally means that tumours occupy a dominant position in their environment through the activation of GLUT1 [188,377], thereby increasing glucose absorption and MCT4-mediated lactate release, finally remodeling tumour metabolism [187,188,378] and promoting tumour progression and invasion [379,380]. Therefore, it is not difficult to explain why NK cells can play a role in the initial stage but lose their role in the promotion and progression of tumour development (Figure 5).

The above interaction process between immunity and tumours makes it easy to see that immune cells play an inhibitory role in tumours and are effective in the early stage of tumour development. However, with the development of tumours, many immunosuppressive molecules are released to induce the differentiation of immune cells and inhibit their anticancer effects. Among these, the accumulation of lactic acid produced by metabolism plays an important role. Lactate is involved in glycolysis, an important component of the tumour microenvironment. Higher lactate levels can be found in advanced tumours and patients with poor prognoses [129,130]. Excessive lactic acid can inhibit the differentiation and maturation of T cells, DC and NK cells [381,382], and cytokine production [35,382]. However, lactate is not only produced as a by-product of glycolysis, nor is it an inhibitory effect on immune cells. It is an important tool for tumours to realize a metabolic adjustment to adapt to the environment and promote tumour progression. In the tumour microenvironment with sufficient nutrition, tumour cells release lactic acid by absorbing glucose, glutamine, etc., whereas in the tumour microenvironment with nutrition deprivation, lactic acid can enter the TCA cycle and be used as the metabolic source of tumour cells for energy supply. Moreover, lactic acid is more important than glucose under these conditions [129]. The accumulation of lactic acid makes the tumour microenvironment to be acidic.

However, in addition to the glycolysis of tumour cells, CAF, Treg, M2 type macrophages, and other immune cells that are “threatened” by tumours are also sources of lactic acid, resulting in an increase in H^+^ in the microenvironment. The increase in H^+^ in the tumour microenvironment during metabolic remodeling and tumour progression is a universal phenomenon, and lactate is an important source [383], which is very important for tumour survival.

The H^+^ is aimed at eliminating the persistent neutralizing stress (such as the neutralization of OH produced by the Fenton reaction and the overproduction of other end products), to maintain homeostasis between synthesis and decomposition (for example, for DNA synthesis and nucleic acid synthesis to facilitate cell division, for attacking immune cells through phospholipid synthesis, and for the synthesis and decomposition cycle through triglycerides) [383]. This was not a hypothesis and was confirmed in subsequent experiments. First, in the differential analysis of the expression profiles in the process of metabolic remodeling, it was found that the signaling pathways affected by metabolic plasticity were related to endoplasmic reticulum (ER) stress genes, and these genes were mainly induced by intracellular and extracellular stress [383]. Second, the acid-base imbalance pressure caused by different pH gradients was examined. Studies have shown that the pH in tumour cells is alkaline (pH > 7.4), while the extracellular pH is acidic (~6.5~−7), which shows a very different pH compared to normal cells (pH ≈ 7.2 intracellular, ≈7.4 in extracellular) [383]. The pH gradient has a dual effect on tumour cells. An intracellular alkaline environment can be conducive to glycolysis to produce more acids and neutralize the OH^−^ produced by the Fenton reaction to avoid the harmful effects of ROS on cells [383,384], and an extracellular acidic environment can inhibit immune response.

Therefore, the significance of tumour cells after metabolic rearrangement is preferentially dealing with stress (such as hypoxia, glucose deprivation, and acid increase) rather than proliferation. However, this increases the heterogeneity of tumour metabolism. Lactic acid causes metabolic symbiosis of the tumour. That is, tumour cell survival in blood vessels rich in oxygen and nutrition can establish a symbiotic metabolic relationship with tumour cells in the hypoxic region. The tumour cells in the oxygen-rich area will absorb the lactic acid produced by hypoxic tumour cells through MCTs [32,33], these tumour cells with rich oxidation and absorption of lactic acid are more likely to use glutamine for TCA cycle metabolism. However, the tumour cells in the hypoxic area are mainly glycolytic, and glutamine utilization is not preferred [36]. This heterogeneous metabolic mode produced by the absorption of lactic acid enables tumour cells to make full use of all microenvironment resources and does not need to compete for resources or interfere with each other. This regulation of metabolic plasticity can reasonably allocate environmental resources to facilitate tumour growth.

### 4.2. Effects of Metabolic Plasticity Regulation on Angiogenesis

Angiogenesis already exists in primary tumour growth, local tumour invasion, and distal metastasis [385], and lactic acid produced by glycolysis plays an important role in angiogenesis [386]. VEGF, fibroblast growth factor 2 (bFGF), and HIF-1 are involved in the process of angiogenesis [387,388,389], while lactic acid can inhibit prolyl hydroxylase-domain 2 (PHD2) and stabilize HIF-1α expression. In non-malignant epithelial cells, HIF-1α induced by lactate can synergistically upregulate bFGF and VEGFR2, with VEGF upregulated synergistically [33,390,391]. Therefore, the tumour needs to maintain neovascularization by introducing metabolism into glycolysis through regulating metabolic plasticity (Figure 5). Tumour cells try to activate glycolysis to facilitate angiogenesis, such as by inducing HIF-1 expression and increasing GLUT1/GLUT3 to enhance glucose absorption [392,393], inducing key enzymes of glycolysis, inhibiting the conversion of pyruvate to acetyl CoA into the TCA cycle [394,395], promoting LDHA transcription, and converting pyruvate into lactate [396,397].

In addition, HIF-1 can activate mitophagy and prevent glucose and fatty acids from entering oxidative metabolism [398,399]. Therefore, considering that the induction of HIF-1 is more significant under hypoxia, tumour cells in this state are more prone to angiogenesis. Thereafter, vascular bud formation is crucial for angiogenesis. It is guided and initiated by tip endothelial cells and then proliferates using stalk cells to extend the vascular wall. This process is accompanied by activating the VEGF/VEGFR2-signaling pathway in endothelial cells [400,401,402]. Glycolysis can promote endothelial cells to compete for “tip” sites and “stalk” sites [403], this process is accompanied by the overexpression of the glycolytic regulatory protein 6-phosphofructo-2-kinase/fructose-2,6-bisphosphatase 3 (PFKFB3) [403] (Figure 5). The knockdown of PFKFB3 can cause endothelial cells to lose these abilities, mainly by increasing cell adhesion and reducing filopodia formation. An in-depth mechanism revealed that PFKFB3 can drive glycolysis, thereby regulating VE-cadherin adhesion. ATP produced by PFKFB3-driven glycolysis can phosphorylate VE-cadherin before endocytosis [404]. Therefore, when PFKFB3 is deleted, glycolysis does not produce ATP, which blocks the endocytosis of VE-cadherin, causing extensive VE-cadherin on the cell surface and impeding cell movement [405]. From the above process, we can recognize the significance of glycolysis in tumour angiogenesis.

### 4.3. Effects of Metabolic Plasticity Regulation on Tumour Migration and Invasion

The regulatory mode of metabolic plasticity is crucial for the successful migration of tumour cells from proximal to distal regions. Thus, tumour cells can adjust the appropriate metabolic mode to adapt to the new microenvironment at the time of metastasis or after reaching the target organ [406]. In general, tumour cell metastasis to the distal end must proceed through the following stages: moving away from the primary site, invading the surrounding tissues, entering the circulation, planting to the distal end (adhesion), and completing migration [407,408].

Cell motility is the basis of tumour metastasis, and motile cells exhibit higher levels of glycolysis [409,410]. The enhancement of glycolytic flux (increased lactate release, enhanced glucose absorption, low OCR/extracellular acidification rate(ECAR), but high NAD^+^/NADH ratios) activates EMT-related molecules snail family transcriptional repressor 1(SNAI), twist-related protein (TWIST), and zinc finger E-box binding homeobox 1(ZEB1), and stimulates yes1-associated transcriptional regulator/Tafazzin (Yap/TAZ)-signaling to enhance cell movement [411]. Correspondingly, the plasticity of metabolic-to-glycolytic transition in tumour cell metastasis [8,406,412] is accompanied by the loss of mitochondrial function [413,414]. For instance, during the invasion and migration of breast cancer cells, the expression of delta-like non-canonical Notch ligand 2 (DLK2), a member of the epithelial growth factor (EGF) repeat superfamily protein, is regulated to enter the mitochondria and interact with the cox assembly factor 3 (COA3) to reduce its activity, thereby decreasing mitochondrial oxidative metabolism and increasing lactate release and glucose absorption, thus promoting the metabolic transition process from OXPHOS to glycolysis and realizing its metastasis [415].

Before leaving the primary site, cancerous epithelial cells must overcome anoikis, a form of apoptosis caused by leaving support from the extracellular matrix [40,41]. Notably, the induction of glycolysis can also enable tumour cells to shift to functional mitochondrial oxidation in an AMPK-dependent manner, thus leading to Snail-driven metastasis and subsequent anti-anoikis apoptosis [38]. This is a fascinating metabolic regulatory mechanism during cell migration. Tumour cells from the primary site require energy to survive when they transfer, and the glycolytic phenotype is the main metabolic mode in this process [416,417]. Although the energy supply to tumour cells is relatively small, its energy conversion efficiency is 100 times that of OXPHOS [42]. Currently, the transferred tumour cells need to exchange materials rapidly during migration, rather than OXPHOSS, which is a “slow-paced” energy supply mode. Contrastingly, the activation of AMPK2 induced by glycolysis enhances the expression of sodium-dependent neutral amino acid transporter type 2 (SLC1A5), a protein that transports glutamine into the TCA cycle for metabolism and increases mitochondrial oxidative stress. The resulting imbalance of redox balance is alleviated by the activation of Nrf2 by AMPK to maintain the energy balance and redox homeostasis of tumour cells and to avoid anoikis. Therefore, some researchers have proposed that glutamine metabolism occurs more easily in aggressive cancer cell lines (owing to stronger metastasis and higher survivability) [418,419,420], and that migrating and invading cells use more glutamine for metabolism [421,422].

Cell metastasis dominated by glutamine metabolism often occurs when the source of cell energy supply substrate (mainly glucose) is reduced, whereas tumour cells in remote hypoxia mainly rely on glucose for metabolism, and lactate is secreted through MCT4 and then taken up by tumour cells in oxidative metabolism through MCT1 [33]. However, tumour cells with the oxidative metabolic phenotype of lactate are more likely to undergo the TCA cycle by absorbing glutamine, which is inconsistent with the nutrient source (mainly glucose) taken up by cells undergoing glycolysis [36]. This metabolic symbiosis process (tumour cells prefer glycolysis to absorb glucose to produce lactic acid, and tumour cells prefer mitochondrial oxidative phosphorylation that absorbs lactic acid to metabolise glutamine) also fully reflects that tumour cells with metabolic heterogeneity can be more flexible in using environmental resources for survival. Notably, when glucose starvation occurs, tumour cells enhance the absorption of glutamine and, at the same time, upregulate matrix metallopeptidase 9 (MMP9) to enhance the metastasis of tumour cells and make them leave the harsh environment faster [423]. Therefore, the effect of glutamine on metastasis is not caused by its activation of OXPHOS, and the migration mechanism of glutamine to tumour cells is mainly realized by activating EMT [424,425]. EMT causes epithelial cells to lose polarization and cell–cell adhesion, resulting in a mesenchymal phenotype that can migrate and invade. Meaningfully, compared with the early progression stage in the 20 tumour types, when tumour cells developed, the upregulation of the EMT process was accompanied by the downregulation of mitochondrial-related genes [426]. Correspondingly, the molecules related to the glycolytic phenotype were significantly upregulated in interstitial cells [427], and key molecules that affect glycolysis, such as HIF-1α, PKM2, and HK2, can directly regulate the expression of EMT protein, such as E-cadherin, vimentin, and Twist1 [39,428]. Therefore, the increase in cell migration may be due to the interaction between glycolysis flux and EMT. However, we need to identify the factors that trigger enhanced mitochondrial metabolism (such as absorption of glutamine and activation of AMPK) also exist, whose purpose is to avoid excessive tumour cell death events (such as anoikis) during migration. The regulation of metabolic plasticity at this time is not only for cell energy supply. The by-product of glycolysis activates HIF-1α and promotes the upregulation of VEGF, bFGF, and their receptors. Thus, the adaptive expression of genes enables the tumour to adapt to the new environment faster when it migrates [429]. Therefore, during tumour cell transfer from the primary site to the secondary region, the plastic regulation of energy metabolism is not only a single stimulus but is a collaborative operation (Figure 6).

Therefore, glycolysis may be the main metabolic mode that enhances cell migration. In contrast, tumour cells with glycolytic phenotype can enhance mitochondrial metabolism (such as the absorption of glutamine and activation of AMPK); its real purpose is to avoid excessive tumour cell death events during migration (such as anti-anoikis by activation of AMPK). These glycolytic tumour cells are more aggressive (owing to stronger metastasis and higher survivability) [418,419,420] with more glutamine for metabolism [421,422]. Therefore, there may be at least two types of metastatic tumour cells, according to metabolic heterogeneity. One is glycolytic tumour cells with glucose for metabolism, and with reduced glucose, hypoxia, and/or high LA, these cells can develop into glycolytic tumour cells with glutamine for metabolism; these cells were more aggressive (Figure 6). Therefore, regulation of metabolic plasticity at this time is not only for cell energy supply but also as a by-product of glycolysis with multiple functions. The adaptive expression of genes enables the tumour to adapt to the new environment faster when it migrates.

## 5. Conclusions

Plasticity is the most prominent characteristic of tumourigenesis. It is also an advantageous biological that helps tumour cells adapt to environmental changes. Metabolic plasticity is an essential component of cellular plasticity, of which glycolysis and mitochondrial oxidative phosphorylation are two important components. Gene or genome instability caused by various factors initiates tumourigenesis and is accompanied by glycolysis and mitochondrial oxidative phosphorylation, enabling tumour cells to “arbitrarily” adjust their metabolic patterns. This change in metabolic mode also enables the daughter cells to survive better in “residential areas” with different nutrients, oxygen stresses, and immune cells by regulating metabolic plasticity according to the characteristics of the environment. Moreover, regulating metabolic plasticity partly prevents cell death and decreases viability caused by their respective metabolic defects. Therefore, regulating metabolic plasticity is a compensation for energy and a mechanism for adaptation and survival by optimizing a series of physiological, biochemical, and molecular regulatory processes. Therefore, to achieve an effective treatment of tumours, especially the occurrence and development of tumours through metabolism, multiple intervention strategies, which can block the metabolic changes generated by tumour cells, should be adopted. As tumourigenesis and its progression can occur and be regulated in multiple dimensions, more comprehensive metabolomic profiling and characteristic analyses on different stages and types of tumour cells are warranted in future studies.

## Figures and Tables

**Figure 1 ijms-24-07076-f001:**
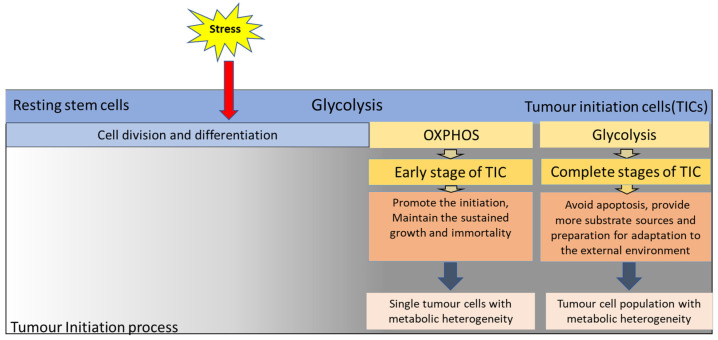
Influence of metabolic plasticity regulation on tumour initiation. Compared with other periods, stem cells at the resting stage (resting stem cells) are more susceptible to external stimuli during division and differentiation, resulting in increased gene instability, tumorigenesis initiation, and appearance of TICs. At the same time, mitochondrial OXPHOS may exist in the early stage, that is, for the immortalization and further transformation of TIC by promoting aspartate synthesis and subsequent nucleotide metabolism and other biochemical reactions. Glycolysis has essential roles during tumour initiation and affects the entire TIC stage, which is necessary for tumour cell initiation. These processes generate at least two types of metabolic heterogeneity; the metabolic plasticity originates from a single tumour cell and is caused by the heterogeneity of the tumour cell population, which helps tumour cells cope with complex internal and external microenvironments.

**Figure 2 ijms-24-07076-f002:**
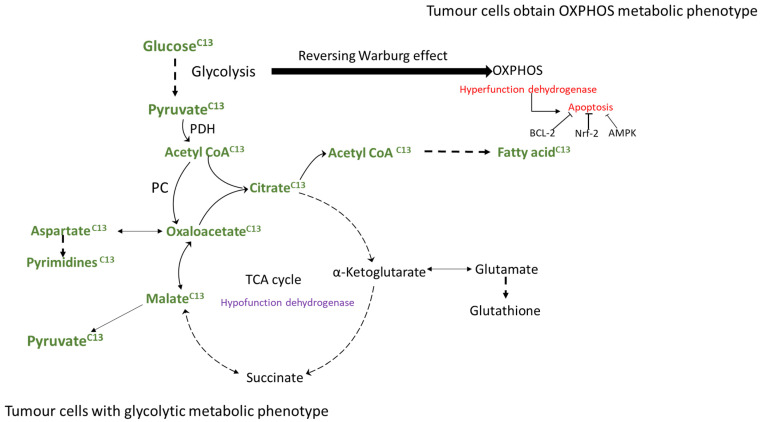
Tumour cells with glycolytic phenotype have high mitochondrial oxidative metabolism and can switch to OXPHOS. Glycolytic tumour cells absorb glucose (C13 labelled) and produce pyruvate. Pyruvate can be catalysed by PC and PDH for producing carbon-derived metabolites (such as citrate and oxaloacetate). Tumour cells that perform this process are “hypofunctive on dehydrogenase”. The TCA cycle in this process can be used for biosynthesis (such as synthesizing fatty acid, pyrimidines, and pyruvate). Switching from glycolysis to OXPHOS can cause a reverse Warburg Effect and change the phenotype of tumour metabolism. The hyperfunction successful dehydrogenase activated by OXPHOS function can generate more oxidative stress, causing cell death, which is not beneficial for the long-term survival of tumour cells. Tumour cells can then counter the endogenous damage caused by increasing the anti-apoptotic proteins such as B-cell lymphoma-2 (BCL-2), nuclear factor erythroid 2-related factor 2(Nrf-2), and 5′-AMP-activated protein kinase (AMPK). Tumour cells that perform this process are “hyperfunctive on dehydrogenase”.

**Figure 3 ijms-24-07076-f003:**
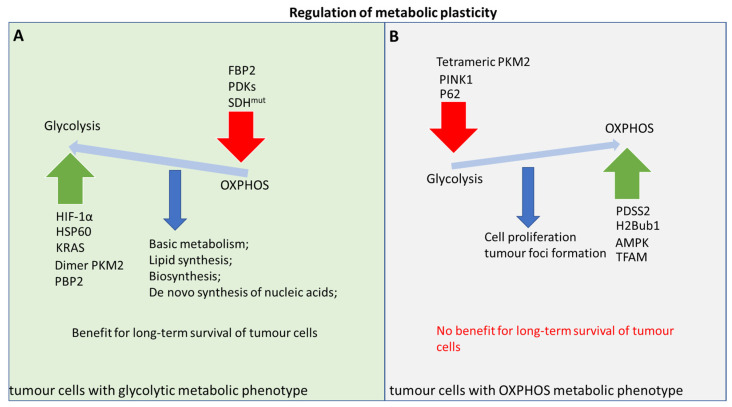
Partial metabolic plasticity regulation in tumour cells. Many molecules can cause tumour cells to switch between glycolysis and OXPHOS phenotype. (**A**) For tumour cells with glycolysis metabolic phenotype, many factors can directly or indirectly regulate the glycolysis process, such as HIF-1α, which can regulate HK-2, PDK1, GLUT1; HSP60, which can strengthen glycolysis reaction, reduce OXPHOS, and reduce the transition from glycolysis to OXPHOS; KRAS, which can regulate the critical enzymes of HK, PFK1, LDHA, and GLUT1, thus affecting the glycolysis process; inactivated PKM2 (Dimer PKM2), which manifest that metabolites accumulate in tumour cells mainly by glycolysis; and FBP2, which can bind to and activate PKM2, making cells for executing glycolysis reactions. In addition, by affecting the mitochondria-related molecules, which can also shift the metabolism to glycolysis phenotype, such as FBP2, it can present the inhibitory effect on mitochondrial transcription factor NRFs and TFAM. Some PDK isoforms can restrict pyruvate entry into mitochondria, and the mutation of SDH can promote the expression of multiple glycolytic enzyme genes. High glycolysis flux levels can help basic metabolism, lipid metabolism, biosynthesis, and denovo synthesis of nucleic acid. In addition, glycolysis flux does not produce oxidative stress molecules to harm tumour cells. Therefore, the glycolysis metabolic phenotype is beneficial for the long-term survival of tumour cells. (**B**) Tumour cells with OXPHOS metabolic phenotype also have many factors that can directly or indirectly regulate the OXPHOS process. For example, activating PKM2 (Tetrameric PKM2) can catalyze phosphoenolpyruvate (PEP) to pyruvate and promote its entry into the TCA cycle. An overexpression of PTEN-induced kinase 1 (PINK1) can promote mitophagy, reduce glycolysis, and increase OXPHOS. Autophagy protein P62 can recognize polyubiquitination of HK-2, catalysed by E3 ligase TRAF6, and then initiate the degradation of HK-2, thereby inhibiting the glycolysis levels. In addition, some effector molecules can directly promote OXPHOS, such as by directly regulating PDSS2, increasing the activity of CoQ10 and mitochondrial Complex I. H2Bub1 regulates a series of mitochondrial genes that can be used for OXPHOS and inhibit the glycolysis level. AMPK is a core molecule regulating mitochondrial energy metabolism, which can enhance mitochondrial metabolism in multiple ways. TFAM plays a critical role in mitochondrial activity and gene transcription. High levels of OXPHOS can help with proliferation and tumour foci formation. However, OXPHOS is not good for the long-term survival of tumour cells because excessive OXPHOS may lead to more oxidative stress molecules that can damage tumour cells and cause the death of tumour cells.

**Figure 4 ijms-24-07076-f004:**
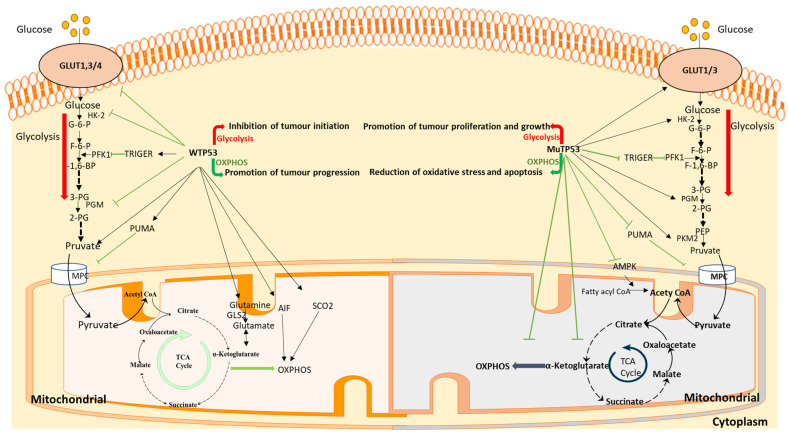
Effects of WTP53 and mutation of MuTP53 on tumour metabolism regulation. P53 can affect the metabolic plasticity process in multiple ways, thus affecting cancer pathogenesis and development. For the positive response to mitochondrial metabolism, WTP53 can promote the transcriptional level of GLS2 to catalyse the conversion of glutamine to glutamate, which is the source material of the TCA cycle. In addition, WTP53 can maintain the expression of apoptosis-induced factor (AIF), while SCO2, AIF, and SCO2 are required for mitochondrial cytochrome c oxidase assembly and maintain the electron transport chain. In contrast, for the negative response to glycolysis, WTP53 can directly regulate the first step of glycolysis by downregulating the transcriptional level of GLUT1/4 or suppressing NF-κB that can activate GLUT3. Moreover, the TIGAR can be encoded by WTP53 before hydrolysing and reducing the expression level of PFK1, which is the rate-limiting enzyme of glycolysis. However, in liver cancer cells, WTP53 can disrupt the binding of MPC to pyruvate by inducing PUMA and inhibit the uptake of pyruvate from mitochondria, thus restraining OXPHOS. These data indicate that P53 may promote tumour progression but prevent cell carcinogenesis. In contrast with WTP53, MuTP53 may have opposite effects on tumour metabolism regulation. For glycolysis, MuTP53 can promote multiple glycolytic enzymes expression through different molecular pathways, such as HK-2, PFK-1, phosphoglucomutase (PGM), and PKM2. In contrast, for the mitochondrial metabolism, MuTP53 can inhibit the activity of α-KGDC and COXIV, then reduce the level of OXPHOS. In short, WTP53 or MuTP53 will have a complete effect on the type of energy metabolism.

**Figure 5 ijms-24-07076-f005:**
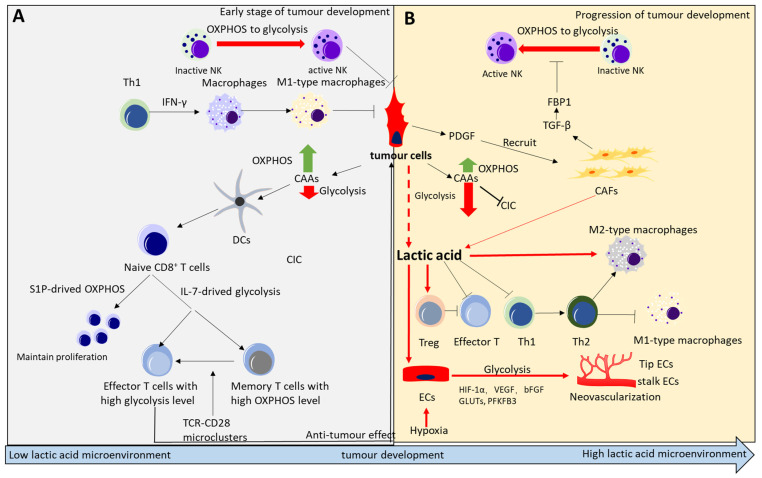
Effects of metabolic plasticity regulation on tumour microenvironment. The stages of tumour development are also per stage of adapting and transforming for the microenvironment. (**A**) At the early stage of tumour development, CAAs produced by the death of tumour cells can be taken up by DCs, and CAAs are transmitted to the naive CD8^+^ T cells and differentiate into effector CD8^+^ T cells, then recognize the tumour cells and initiate the death of the tumour. This repetitive process is called CIC. The uptake of glucose, glutamine, and other nutrients can be utilized by glycolysis and OXPHOS. At the same time, OXPHOS can produce ROS, causing more damage to the tumour cells and then inducing DNA mutations, which increase its immunogenicity. Moreover, S1P and IL-7 can influence the differentiation of naive CD8^+^ T cells by S1P-driving OXPHOS and IL-7-driving glycolysis. Differentiated memory CD8^+^ T cells have better mitochondrial activity, and upon TCR and CD28 stimulation, their glycolytic metabolism can be promoted by activating the PI3K-mTOR-signaling pathway and differentiating into effector T cells. In addition, IFN-γ secreted by Th1 can activate macrophages to develop into M1-type macrophages but is unstable in an acidic microenvironment. NK cells can kill target cells directly, which is closely related to metabolic changes such as the activation of NK cells needed to switch its metabolism from mitochondrial oxidation to glycolysis. (**B**) In the progression of tumour development, the microenvironment favourable for the tumour was formed. The transition of OXPHOS to glycolysis or lactic acid can reduce cells’ immunogenicity, thus inhibiting the CIC. Excessive lactic acid (LA) produced by glycolysis is the inhibitory molecule of all anti-tumour immune cells, which can be absorbed by Treg cells, then suppressing effector CD8^+^ T cells. An acidic microenvironment caused by LA can make IFN-γ unstable. Thus, it inhibits the differentiation of Th1 cells and turns to Th2, which antagonizes the differentiation of M1-type macrophages. Except for tumour cells, CAFs are the main source of lactic acid, which is recruited by platelet-derived growth factor (PDGF) secreted through the tumour cells. Moreover, CAFs are the origin of TGF- β, which can induce FBP1 expression. This weakened the glycolytic level of NK cells, thereby inactivating NK cells. Angiogenesis is an important characteristic of tumour cells, and glycolysis contributes to this process. Lactic acid and hypoxia can promote this process by stabilizing HIF-1, VEGF, bFGF, VEGFR2, etc. The formation of vascular buds is crucial in the process of angiogenesis, which is guided and initiated by tip endothelial cells and stalk cells. Glycolysis can promote endothelial cells to compete for “tip” sites and “stalk” sites, and this process is accompanied by the overexpression of the glycolytic regulatory PFKFB3.

**Figure 6 ijms-24-07076-f006:**
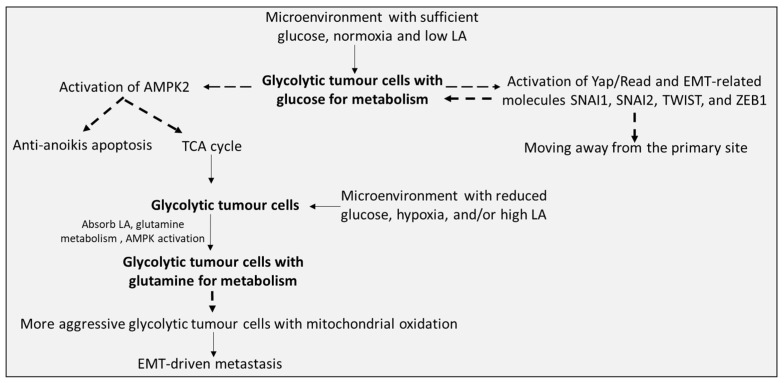
Effects of metabolic regulation on tumour metastasis. Glycolysis and Mitochondrial oxidative metabolism jointly contribute to the invasion and migration of tumour cells. Under the condition of sufficient glucose, normoxia, and low LA, glycolytic tumour cells can enable tumour cells to shift to functional mitochondrial oxidation in a manner of activation dependent on AMPK, thus leading to anti-anoikis apoptosis. In addition, glycolytic flux can activate EMT-related molecules SNAI1, SNAI2, TWIST, and ZEB1, and stimulate Yap/YAZ-signaling pathway to drive cell movement. With the starvation of glucose, hypoxia, and increased lactate acid, glycolytic tumour cells can absorb LA and use glutamine for mitochondrial oxidative metabolism. AMPK activated by glycolysis also can enable tumour cells to shift to functional mitochondrial oxidation. The glycolytic tumour cells with glutamine for metabolism showed more aggression, and together with glycolytic tumour cells with glucose for metabolism, both can activate EMT to drive cell metastasis. The dotted arrows represent that there is a multi-step regulatory pathway between the upstream and downstream of molecular events, which is a brief description rather than one-step regulation.

**Table 1 ijms-24-07076-t001:** Drug therapy for key enzymes that can cause metabolic remodeling.

Drug	Medication Conditions	Mechanism
Cisplatin	HK2 shRNA	Deletion or decrease of HK-2 could produce more ROS, pro-apoptotic molecules, and mitochondrial dysfunction, reduce the drug resistance of tumour cells [154,155,156].
Metformin + sorafenib	HK2 deletion	
Clotrimazole	VDAC disruption	The use of PFK inhibitor clotrimazole can reduce glycolysis under the condition of VDAC disruption, prevent the metabolic rearrangement process, and inhibit the growth and invasion of tumour cells [157].
HrDNase1	DNASE1L3 low expression	HrDNase1 could increase the level of DNASE1L3, then activate the P53-apoptosis pathway, thus inhibiting tumour growth, invasion, and migration [169].
Cetuximab	TRAP1 inhibition	The inhibition of TRAP1 could reduce its binding with PFK1, then trigger the degradation of PFK1, enhancing the sensitivity of CRCs to cetuximab [163].
DASA + metformin	non-sugar environment	DASA inhibits the invasion of tumour cells caused by PKM2 nuclear localisation, and metformin inhibits COXI activity of tumour cells, reduces the growth and promotion of tumour cells [187].
MLN4924 + shikonin + metformin	Tumour cells with high levels of both OXPHOS and glycolysis	MLN4924 inhibits TCA level, shikonin inhibits PKM2 mediated glycolysis, and metformin was used for drug intervention [175,189].
MI-D	DMEM GAL medium	Citric acid could decrease the level of anti-apoptotic molecule expression and increase the level of apoptosis, initiating and executing molecule expression. Then, tumour cells become more sensitive to anti-tumour drugs [210,212,218].
Citric acid + 3-bromo-pyruvate, cisplatin, or celecoxib	—	
Doxorubicin + Sorafenib	Insensitive to drugs for mutant IDH1	The risk scores of glycolysis-related genes were correlated with those two drugs and had better sensitivity [268].
Normoxia + radiotherapy and chemotherapy	tumour cells have the potential of OXPHOS and glycolysis	Remove the cells from the hypoxic area, and then use the corresponding radiotherapy and chemotherapy methods, which can produce more ROS [274,279].
BAY 87-2243(B87) + DMKG	B87 alone is not very effective in inducing tumour cell death	DMKG can produce a strong tumour-killing effect together with B87 [274].
JHU-083	—	Inhibition of α-KG [276].

**Table 2 ijms-24-07076-t002:** Partial molecules that influence Metabolic plasticity process.

Metabolic Type	Factors that Affecting Metabolic Plasticity	Tumour Cell Types	Significance
From Glycolysis to OXPHOS	Resting stem cells are more susceptible to external stimuli during division and differentiation, resulting in increased gene instability, tumorigenesis initiation, and the appearance of tumour-initiating cells (TICs)	The early stage of TIC	For the immortalization and further transformation of TIC by promoting aspartate synthesis and subsequent nucleotide metabolism and other biochemical reactions [83,84,85,86].
Be inclined to Glycolysis		The complete stage of TIC	Avoid apoptosis, and provide more substrate sources and preparation for adaption to the external environment [58,59].
OXPHOS	With sufficient oxygen	Glioma stem cells (CSCs)	To adapt to different microenvironments and meet their vigorous growth and proliferation [117,118,119,120].
Be inclined to Glycolysis	In the hypoxic regions		
Be inclined to OXPHOS	HIF-1^low^/pAMPK^high^	Liver hepatocellular carcinoma (HCC), lung adenocarcinoma (LUAD), breast invasive carcinoma, stomach adenocarcinoma, acute myeloid leukaemia and pancreatic adenocarcinoma (PAC)	Hybrid phenotype contributes to metabolic plasticity, allowing tumour cells to adapt to various microenvironments [136].
Be inclined to Glycolysis	HIF-1^high^/pAMPK^low^		
Hybrid phenotype	HIF-1^high^/pAMPK^high^		
From Glycolysis to OXPHOS	Deletion of HK-2	Retinal cells, Gastric Cancer, glioblastoma multiforme	More ROS and apoptosis-promoting molecules produce [142,143,144,145].
From Glycolysis to OXPHOS	Inhibitory of PFK expression	Pancreatic ductal adenocarcinoma (PDAC), non-small cell lung cancer (NSCLC) cells, and colorectal cancer (CRCs)	Provide energy for the vigorous metabolism of tumour cells [157,158,159,162,163].
Be inclined to OXPHOS	Dimer PKM2	-	Entering the nucleus to participate in the transcription and regulation of many molecules, affecting the process of tumour proliferation, migration, angiogenesis, and death [180,181,182].
Be inclined to Glycolysis	Tetramer PKM2		Providing energy for tumour growth and proliferation [176,177].
Be inclined to OXPHOS	Citrate synthase (CS)	-	Increase the burden of mitochondrial, causing apoptosis [208,209,210,211,212,213,214].
Be inclined to OXPHOS	Mutation of Isocitrate Dehydrogenase (IDH)	-	Make the (R) enantiomer of 2-hydroxyglutarate (R)-2HG) generate and release abnormal metabolism of lactate and pyruvic acid [248,249,250,251,252,253,254].
Be inclined to OXPHOS	High expression of FBP	-	Inhibit HIF-1α, interfere with glycolysis, and hinder tumour cell proliferation, invasion, and migration [307].
Be inclined to Glycolysis	FBP2 located in the nucleus with c-Myc	-	limiting the expression of mitochondrial genes and reducing OXPHOS energy supply [308].
Be inclined to Glycolysis	Mutations in succinate dehydrogenase (SDH) and fumarate hydratase (FH)	Hereditary paragangliomas, leiomyomatosis and kidney cancers	Reducing the degradation of HIF and promoting the expression of multiple glycolytic enzyme genes [309,310,311].
From OXPHOS to Glycolysis	inhibition of HSP60	renal cell carcinoma	Enhance de novo nucleotide synthesis and lipid synthesis [312].
From Glycolysis to OXPHOS	Directly regulating PDSS2	hepatoma cells	Inhibit foci formation, colony formation in soft agar, and tumour formation in nude mice [313].
OXPHOS	WTP53	-	Reduce harmful products, such as ROS, and obtain more stable genomic DNA, thus avoiding the initiation of tumorigenesis [318,319,320,321,322,323].
From OXPHOS to Glycolysis	MuTP53		Promotion of tumour proliferation and growth [319].

## Data Availability

Data available in a publicly accessible repository.

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
