# Peer review of "Regulative Roles of Metabolic Plasticity Caused by Mitochondrial Oxidative Phosphorylation and Glycolysis on the Initiation and Progression of Tumorigenesis"

_ijms, 2023, doi:10.3390/ijms24087076_

Round 1
Reviewer 1 Report (Previous Reviewer 1)
The paper of Niu et al. is devoted to a very hot topic on metabolic rewiring and plasticity in cancer. This is a comprehensive review summarizing and discussing a lot of important information on different aspects of metabolic plasticity.
However, a number of points should be addressed:
The authors should further break the text into paragraphs for logical structuring and ease of perception of the material
24…25 “. In the presence of oxygen, the metabolites are transmitted through respiratory chain enzyme complexes I-IV” - not exactly, indeed, respiratory chain transport electrons (from metabolites - NADH and succinate) to molecular oxygen
191…” They also did not completely exclude the high consumption of GAPDH, PGK, and LDH.” What do authors mean? What is “consumption”? GAPDH, PGK, and LDH – are enzymes, not metabolites
The authors have provided a lot of interesting and valuable information in the text about examples of metabolic switch between glycolysis and oxphos, as well as how either glycolysis and oxphos may help to cancer stem and non-stem cells to adopt to various conditions dependent on the situations. I suggest authors to summarize this valuable information in the table format to simplify the perception and make their review unique.
The meaning of Figure 1 is completely unclear. In my opinion, it does not match the description, although the authors provide excellent examples of metabolic plasticity in the text.
Figure 2 and “hypofunction and hyperfunction dehydrogenase”. The meaning of these is tentively clear. However, are they two new terms? If the authors have formulated these concepts, they should write, for example: «We called these cell “hypofunctive on dehydrogenase”»
372..373 “In contrast to HK, which is the most important rate-limiting enzyme of glycolysis, targeted drugs acting solely on A may have the same anti-cancer effect.” What is “A”, what do authors mean?
413…452 The main properties of PKM2 int the regulation of glycolysis and oxphos should be briefly described (dimer or tetramer different functions), for instance, see Zahra, K., Dey, T., Mishra, S. P., & Pandey, U. (2020). Pyruvate kinase M2 and cancer: the role of PKM2 in promoting tumorigenesis. Frontiers in oncology, 10, 159.
455..456 “It metabolises oxygen molecules to ATP for energy supply through the electron transfer chain located in the mitochondrial inner membrane and ETC complex (I, II, III, IV)” – not true statement
Figure 4. Authors should separate the effects of WT_p53 and mutant p53 in figure
Author Response
We thank you for allowing us to revise our manuscript and appreciate your positive and constructive comments and suggestions on our manuscript entitled “Regulative roles of metabolic plasticity caused by mitochondrial oxidative phosphorylation and glycolysis on the initiation and progression of tumorigenesis(ijms-2228559)”.
We have tried our best to revise our manuscript according to the comments. Please find the revised manuscript attached, which we would like to resubmit for your kind consideration.
The main corrections in the paper and the responses to your comments are as follows:

Reviewer 2 Report (Previous Reviewer 2)
Dear authors,
I have analyzed the revised manuscript but unfortunately the revision has not solved the major concerns that prevent the work for being sufficiently qualified for publication is a scientific journal. English language and the concepts proposed are oversimplified. There are lots of already published papers that properly discuss the role of metabolic plasticity in cancer.
Author Response
Response to the reviewer’s comments:
Dear reviewer:
We apologize for the poor language and concepts of our manuscript. We worked on the manuscript for a long time, and the repeated addition and removal of sentences and sections led to poor readability. We have now worked on both language and readability and have also involved native English speakers for language corrections with the help of English editing services. Please see the attachment for the certificate confirmation. We hope that the flow and language level have been substantially improved. We have tried our best to improve the manuscript and made some changes. However, these changes will not influence the content and framework of the paper.
We appreciate the reviewer’s earnest and warm work and hope the correction will be approved. Once again, thank you very much for your comments and suggestions. Looking forward to hearing from you.
Thank you and best regards.
Yours sincerely,
Nan Niu

Reviewer 3 Report (New Reviewer)
The review "Regulative roles of metabolic plasticity caused by mitochondrial oxidative phosphorylation and glycolysis on the initiation and progression of tumorigenesis" by Nan Niu et al. provides a comprehensive review on the regulation and maintaining the balance between mitochondrial oxidative phosphorylation and glycolysis in cancer cells. The authors have done a good job of synthesising recent research on the subject and highlighting the importance of metabolic plasticity in tumour development.
The review is well-structured and provides the readers with very detailed information about the topic, focusing on elucidating the characteristics of metabolic plasticity in tumour progression. The authors have presented the information in a systematic manner; the six figures included in the article are clear and correlate well with the text, enhancing the reader's understanding of the subject matter. However, the review might be challenging for readers without expertise in the field, as it contains in-depth analysis and expert terminology that may be difficult for non-experts to follow.
The referencing in the article is correct and extensive, with a minimum of self-citation. The majority of references are no more than 10 years old and the review contains many very recent publications (no more than 5 years old), demonstrating the authors' commitment to keeping up with the latest research in the field.
Overall, the authors have made a valuable contribution to the literature on the role of metabolic plasticity in tumour progression. I believe that the review will be a useful resource for researchers, clinicians and students interested in the field of cancer biology and will serve as a useful reference for future research in this area.
In conclusion, I recommend that the paper be published in IJMS after minor revisions. I would just like to make two minor comments:
1) The authors use both "tumour" and "tumor" throughout the manuscript. Please check and standardise this word.
2) Please pay attention to the text formatting - there are often missing spaces between a word and a reference/figure bracket and similar typos.
Author Response
Thank you for your positive and constructive comments and suggestions on our manuscript entitled “Regulative roles of metabolic plasticity caused by mitochondrial oxidative phosphorylation and glycolysis on the initiation and progression of tumorigenesis(ijms-2228559)”. We would like to express our great appreciation for your comments on our paper.
We have tried our best to improve the manuscript and made some changes in the manuscript. However, these changes will not influence the content and framework of the paper. Please find the revised manuscript attached, which we would like to submit for your kind consideration. The main corrections in the paper and the responses to your comments are as follows:

Reviewer 4 Report (New Reviewer)
In the current review, Niu et al. elucidate the importance of metabolic plasticity in cancer cells. Metabolic plasticity is a characteristic of cancer cells that makes them able to switch their metabolic phenotype from glycolysis to OXPHOS and viceversa in different conditions. Firstly, authors discuss the role of metabolic plasticity in the initiation of tumorigenesis, in particular regarding the transformation of stem cells. Then, authors describe the effect of molecules and enzymes on metabolic plasticity. Finally, authors underline the importance of metabolic plasticity in tumor microenvironment that is characterized by the presence of immune cells, a high angiogenesis and in tumor invasiveness. Overall, the topic of the review is interesting and very deepen, although some concepts are redundant and unclear. In my opinion the review could be published after making major revision and checking English language. I have some comments:
- Pay attention to the use of abbreviations because there are many abbreviation not explained in the text and so the reading became difficult. For example Line 58: OXPHOS; line 74-75: GLUT3, EGFR, MAPK, Akt; line 93: EMT; line 133: MEIS1, HIF-1α; line 136: PDK; line 145: c-Myc, NF-kB; line 147: ESC; line 151: KRAS; line 155: HK, PFK1, LDHA, GLUT1; line 174: TFAM; line 186: GSCs, ROS; line 191: GAPDH, PGK, LDH; line 225: TIC; line 272: TCA; line 437: DASA; line 457: ETC; line 650: NRFs, TFAM; line 715: PUMA; line 717: HCC; line 781: TCR, CD28; line 903: MCTs; … I suggest to use the abbreviation for the first time between parenthesis and then only the abbreviation or use abbreviations and add a paragraph with all the abbreviation legends.
- Line 120: “In the advanced stage, tumour stem cells can undergo asymmetric division at the self-renewal stage to generate daughter cells with different phenotypes, which may illustrate some puzzle, such as whether the metabolic characteristics are displayed by a single tumour cell or whether the hybrid characteristics are due to the heterogeneity of tumour cell populations with different metabolic modes.”
Line 213: “And many studies indicated that in the whole stage of initiation of tumorigenesis, activation of the glycolytic pathway maybe not only compensate for energy shortage caused by the loss of mitochondrial function, the whole stage of tumour initiation driven by excessive glycolysis flux can also avoid the generation of apoptosis induced by ROS produced by OXPHOS[100, 104, 105].”
Line 1002: “Therefore, from glycolysis promoted cell migration to anoikis caused by leaving the primary site, by absorbing glutamine to absorption of glutamine causes EMT expression, then to reconstitution into enhanced glycolytic flux and decreased mitochondrial function, glycolysis is the main metabolic mode that enhances cell migration”
Line 1026: “Therefore, from the process that glycolysis promoted cell migration, to anoikis caused by leaving the primary site, to anti-anoikis by absorbing glutamine to absorption of glutamine causes EMT expression, then to reconstitution into enhanced glycolytic flux and decreased mitochondrial function, glycolysis is the main metabolic mode that enhances cell migration.” The reading is difficult, so could you reformulate these sentences?
- Legend of Figure 1: insert “stress” between parenthesis as an example of external stimuli. In the legend OXPHOS is described and then glycolysis, please invert the description following the text in the figure.
- Line 267: “For instance, some studies have reported that lactate dehydrogenase(LDH) is upregulated after cell cancerisation, and following this concept, the upregulation of LDH may be more inclined to the increasing level of glycolysis, and related research and treatment will also be inclined to it.” Please, add reference.
- Please, enlarge Figure 2 and 5.
- Line 606: change PH with pH.
- Table 1: Could you add a reference for each mechanisms? In addition, could you leave a space between one sentence and another?
- Figure 3: Could you insert panel A (for tumor cells with glycolytic metabolic phenotype) and panel B (for tumor cells with OXPHOS metabolic phenotype) to better clarify the legend? In addition, I suggest to delete numbers in the first part of the legend or add other numbers in the second part to list all the factors.
- Figure 5: Could you insert panel A (for early stage of tumor development) and panel B (for progression of tumor development) to better clarify the legend?
- Line 889: “Studies have shown that the pH in tumour cells is alkaline (pH > 7.4), while the extracellular pH is acidic (≈ 6.5 ~ 7).” The sentence is repeated twice.
- Figure 6: The legend is full of information that are not presented in the figure. Please, complete the figure.
Author Response
We thank you for allowing us to revise our manuscript and appreciate your positive and constructive comments and suggestions on our manuscript entitled “Regulative roles of metabolic plasticity caused by mitochondrial oxidative phosphorylation and glycolysis on the initiation and progression of tumorigenesis(ijms-2228559)”. All responses to related questions are listed below.

Reviewer 5 Report (New Reviewer)
This review summarized the metabolic plasticty, especially the detailed characteristic of glycolysis and OXPHOS in tumors.
1. Although the metabolism of glycolysis and OXPHOS are the main issues, nearly no evidences of mitochondrial energy metabolsim such as seahorse analysis are mentioned.
2. The molecular mechanisms are fully discussed, but the endpoints or effects are not explained. We are still unknown how these molecules affect the clinical parameters, and it is out of the scope of clinical application, which was claimed in this review.
3. Multi-omics data analysis are suggested to reveal the difference between normal cells and tumors at different stages, rather than biochemical experiments. This is a review, so more comprehensive data are needed.
4. Which is the most important molecular initiation event (MIE) or pathway for the transition from normal to neoplastic cells.
5. Many molecules and regulation networks are presented, but the discussion and summarization showed be improved.
Author Response
Thank you for your decision and constructive comments on our manuscript. We have carefully considered the suggestion of the Reviewer and made the necessary changes. We have tried our best to improve our manuscript. All responses to related questions are listed below.

Round 2
Reviewer 4 Report (New Reviewer)
The authors have improved the English language, making careful editing and rephrasing difficult to read sentences. Also, all abbreviations have been explained and added in a list as suggested. The enlarged figures and updated legends have served to clarify some difficult concepts. Concerning Figure 1, I would choose option 1 proposed by the authors. Option 1 clarifies that stress acts directly on cell division and differentiation. In my opinion, the revised review can be published.
This manuscript is a resubmission of an earlier submission. The following is a list of the peer review reports and author responses from that submission.
Round 1
Reviewer 1 Report
The manuscript of Niu et al. is devoted to metabolic plasticity in neoplastic progression.
This is a good review paper which summarizes a lot of data and promotes the modern and reasonable point of view.
On my opinion, only few minor points should be addressed:
Figure 3, left part. “Basic metabolism” (misspelling)
740… “ directly regulate the expression of EMT protein” What proteins? Give examples, please
The authors mentioned different proteins affecting metabolic rewiring. However, such master regulators of energy metabolism as c-Myc, Hif1a and Akt should be addressed in more details. Especially their role in the regulation of metabolic plasticity is interesting
Authors should add the table describing modern drugs targeting metabolic processes and plasticity which are in preclinical and clinical studies
Reviewer 2 Report
Dear authors,
unfortunately I could not appreciate the content of your proposed review for some major reasons: 1) the English language needs extensive editing and rephrasing; 2) the work has the ambitious to cover a too broad area of investigation (from the different stages of tumor development to the intercellular crosstalk of cancer cells with cells of the microenvironment, to the different tumor types) and the overall structure of the review and the images proposed do not easily convey the message that the title suggests; 3) I find that the review highlights the connections between the diverse metabolic features of cancer cells and tumor characteristics (stages, migration, immune escape, etc.) but does not cover the causal link between the two. Therefore, it remains a correlative description. Moreover, some of the elaborations proposed result oversimplified for lack of proper in-depth analysis.
Reviewer 3 Report
The manuscript by Niu et al. is a review aiming to describe the regulative roles of metabolic plasticity caused by mitochondrial oxidative phosphorylation and glycolysis on the initiation and progression of tumorigenesis.
However there are some serious concerns about the composition and the content of the study which, in my opinion, must be addressed before considering for publication
- The main drawback of the study is the chaotic presentation of described mechanisms, with numerous repetitions of addressed points. What is important, substantive correctness is also questionable.
- The novelty of presented mechanisms is doubtful
- With the exception of the very last paragraphs (4.1.2 and Conclusions) the language needs extensive editing – long, unclear sentences, linguistic errors. It is difficult to access the merit.